# Bregman Centroid Guided Cross-Entropy Method

## Abstract

The Cross-Entropy Method (CEM) is a widely adopted trajectory optimizer in model-based reinforcement learning (MBRL), but its unimodal sampling strategy often leads to premature convergence in multimodal landscapes. In this work, we propose $\mathcal{B}$regman-$\mathcal{C}$entroid Guided CEM ($\mathcal{BC}$-EvoCEM), a lightweight enhancement to ensemble CEM that leverages *Bregman centroids* for principled information aggregation and diversity control. BC-EvoCEM computes a performance-weighted Bregman centroid across CEM workers and updates the least contributing ones by sampling within a trust region around the centroid. Leveraging the duality between Bregman divergences and exponential family distributions, we show that BC-EvoCEM integrates seamlessly into standard CEM pipelines with negligible overhead. Empirical results on synthetic benchmarks, a cluttered navigation task, full MBRL pipelines, and a real-world quadruped robot demonstrate that BC-EvoCEM enhances both convergence and solution quality, providing a simple yet effective upgrade for CEM.

## 1 Introduction

The *Cross–Entropy Method* (CEM) is a derivative–free stochastic optimizer that converts an optimization problem into a sequence of rare event estimation tasks (Rubinstein & Kroese, 2004; De Boer et al., 2005). At each iteration, CEM samples $N$ candidates $\{x_j\}_{j=1}^N$ from a parametric distribution $p_{\theta_t}$, selects top lowest-cost samples as an elite set $\mathcal{E}_t$, and updates the parameters by maximizing the log-likelihood of these elites:

$$\theta_{t+1} \;=\; \arg\max_\theta \sum_{x \in \mathcal{E}_t} \log p_\theta(x), \tag{1}$$

optionally smoothed via exponential averaging for stability. Its reliance solely on cost-based ranking instead of gradient information has made CEM a widely adopted solver for high-dimensional, nonconvex optimization tasks in robotics and control (Pinneri et al., 2021; Kobilarov, 2012; Banks et al., 2020).

In model–based reinforcement learning (MBRL), an agent learns a predictive model of the environment and plans through that model to reduce costly real-world interactions (Ha & Schmidhuber, 2018; Nagabandi et al., 2018; Silver et al., 2017). Stochastic model predictive control (MPC) is a widely used planning strategy in this setting (Williams et al., 2016; Okada & Taniguchi, 2020; Chua et al., 2018; Zhang et al., 2022; Deisenroth & Rasmussen, 2011). At every decision step, MPC solves a finite–horizon trajectory optimization problem, executes only the first action, observes the next state, and replans. The CEM is often chosen as the optimizer within this loop due to its simplicity, reliance solely on cost function evaluations, and robustness to noisy or nonconvex objectives.

Despite these advantages, vanilla CEM suffers from its inherent *mode–seeking* nature: as the elites concentrate, it often collapses the search into a local optimum, which significantly limits the exploration in complex multimodal landscapes typical of RL tasks. Ensemble strategies have been proposed to mitigate this issue by running multiple CEM workers. **Centralized ensembles** merge the elite sets of all workers and fit an explicit mixture model (e.g., commonly a Gaussian mixture (Okada & Taniguchi, 2020)). Although more expressive, they introduce additional hyperparameters (number of components, importance weights) and increase computational cost due to joint expectation maximization (EM) steps. **Decentralized ensembles**

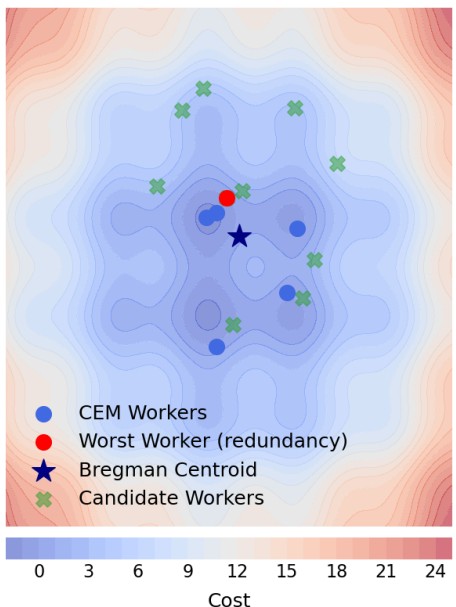 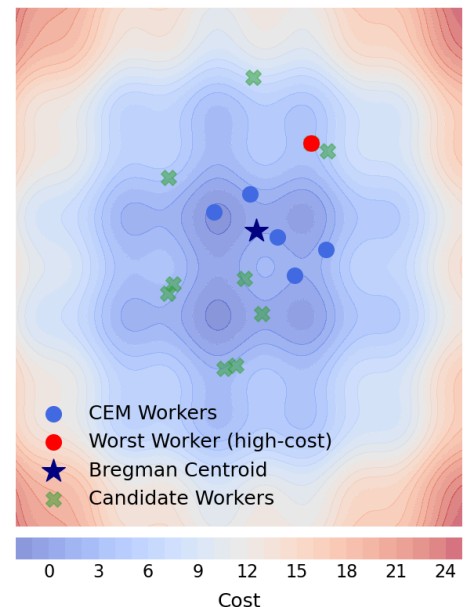

Figure 1: (Revised) $\mathcal{BC}$-EvoCEM working pruning and respawn (means only). Background shows the cost field (cooler = lower cost). • Blue dots are active CEM workers; the star ★ is the Bregman Centroid of these active workers; × are candidate workers proposed near the centroid. **Left:** a *redundant* worker (•) that sits on top of another worker is removed to increase diversity. **Right:** a *poor-quality* worker (•) stuck in a high-cost area is removed. In both cases, the freed budget is reallocated to new candidates around the centroid, focusing search on the promising low-cost basin while keeping coverage diverse.

run multiple CEM instances in parallel, keep them independent, and output the best solution at termination (Zhang et al., 2022). This approach is simple and scalable but tends to duplicate exploration effort and may reach premature consensus if poorly initialized.

**Our Approach.** Motivated by the trade-off between diversity preservation and computational efficiency, we introduce $\mathcal{B}$**regman-$\mathcal{C}$entroid Guided CEM ($\mathcal{BC}$-EvoCEM)**, a hybrid strategy that retains the independent updates of decentralized ensembles yet introduces a simple information–geometric coupling across workers. At each CEM iteration, $\mathcal{BC}$-**EvoCEM** computes a *performance–weighted Bregman centroid* (Nielsen & Nock, 2009) of all workers' distributions. The centroid then defines both a reference point and a *Bregman ball* trust region. Any worker whose distribution lies too close to the centroid or exhibits high cost is respawned by drawing new parameters from this trust region (see Fig. 1).

**Contributions.** 1) We formulate an information–geometric aggregation rule based on Bregman centroids that summarizes ensemble CEM workers with *negligible* computation cost. 2) We provide a lightweight integration into the MPC loop for MBRL, preserving the benefits of $\mathcal{BC}$-**EvoCEM** with the simplicity of a standard warm-start heuristic. 3) Through experiments on multimodal synthetic functions, cluttered navigation tasks, and full MBRL benchmarks, we demonstrate faster convergence with improved performance relative to the vanilla and decentralized CEM.

## 2    (Extended) Preliminaries

**Notation**    We write $\langle \cdot, \cdot \rangle$ for the Euclidean inner product on $\mathbb{R}^d$ and $\| \cdot \|_2$ for the Euclidean norm. For a strictly convex, differentiable function $F : \mathcal{S} \to \mathbb{R}$ on a convex set $\mathcal{S} \subset \mathbb{R}^d$, let $F^*$ denote its convex conjugate and $\nabla F$ its gradient.

**Bregman Divergence and its Centroid.** Given $F$ as above, the Bregman divergence is

$$\mathrm{D}_F(x\|y) \;=\; F(x) - F(y) - \langle x - y, \nabla F(y)\rangle, \qquad x,y \in \mathcal{S}. \tag{2}$$

Although not a metric (it is generally asymmetric and does not satisfy the triangle inequality), $\mathrm{D}_F$ is nonnegative and $\mathrm{D}_F(x\|y) = 0$ iff $x = y$. For the quadratic potential $F(x) = \frac{1}{2}\|x\|_2^2$, one obtains $\mathrm{D}_F(x\|y) = \frac{1}{2}\|x - y\|_2^2$, one half of the squared Euclidean distance. Given a collection of points $\{x_i\}_{i=1}^n \subset \mathcal{S}$, the *Bregman centroid* (right-sided) is the solution to the following minimization problem (Nielsen & Nock, 2009):

$$\boldsymbol{x_c} \;=\; \arg\min_{x \in \mathcal{S}} \frac{1}{n} \sum_{i=1}^n \mathrm{D}_F(x_i\|x). $$

Its minimized value is known as the *Information Radius* (IR) (Csiszár et al., 2004) (*Bregman Information* in (Banerjee et al., 2005)), which characterizes the *diversity* of the set $\{x_i\}$.

**(NEW) Exponential Families and Bregman Divergences.** Many standard distributions are members of a (regular, minimal) *exponential family (EF)*. In canonical form,

$$p_\theta(x) \;=\; h(x)\,\exp\{\langle \theta, T(x)\rangle - \Psi(\theta)\}, \qquad \theta \in \Theta \subset \mathbb{R}^d, \tag{3}$$

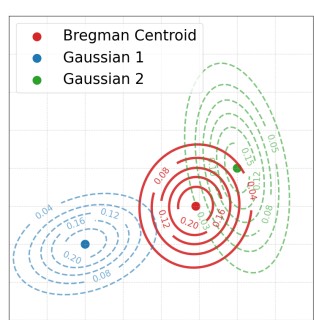

where $\theta$ is the *natural parameter*, $T : \mathcal{X} \to \mathbb{R}^d$ is the sufficient statistic, $h$ is the carrier measure, and $\Psi$ is the convex log-partition (cumulant) function. The *mean/expectation parameter* is

Figure 2: Illustration of the Bregman centroid.

$$\eta \;:=\; \nabla\Psi(\theta) \;=\; \mathbb{E}_\theta[T(X)]. $$

For regular, minimal EFs, $\Theta$ is open and $\Psi$ is strictly convex, so the mapping $\theta \mapsto \eta$ is a bijection onto the *mean-parameter space* $\mathcal{M} := \nabla\Psi(\Theta)$. The Legendre transform yields the conjugate pair: $\eta = \nabla\Psi(\theta)$ and $\theta = \nabla\Psi^*(\eta)$. (See (Barndorff-Nielsen, 2014; Rockafellar, 2015)).

A fundamental result (Theorem 3 of Banerjee et al., 2005) states that for any regular EF with cumulant $\Psi$, there is a *unique* Bregman geometry via $\Psi$ (equivalently via $\Psi^*$). In other words, a regular EF member with a cumulant $\Psi$ is uniquely determined by $D_{\Psi^*}$ in the mean parameter space (equivalently by $\Psi$ in the natural space). We leverage this EF–Bregman duality for principled and efficient aggregation later in the paper.

## 3 (NEW) Problem Statement

**Cross-Entropy Method** Consider black-box minimization of a possibly nonconvex, noisy objective

$$\min_{x \in \mathcal{X}} J(x), \tag{4}$$

where $\mathcal{X} \subset \mathbb{R}^d$ and only pointwise evaluations of $J$ are available. The *Cross–Entropy Method (CEM)* converts the problem in equation 4 into a sequence of parametric rare-event estimation steps (Rubinstein & Kroese, 2004). At iteration $t$, CEM maintains a parametric sampling distribution $p_{\theta_t}$ over $\mathcal{X}$, draws $N$ candidates $\{x_j^{(t)}\}_{j=1}^N \sim p_{\theta_t}$, selects an *elite set* $\mathcal{E}_t$ containing the top (lowest-cost) samples, and updates the parameter $\theta_t$ by maximum likelihood on $\mathcal{E}_t$:

$$\hat{\theta}_{t+1} \;\in\; \arg\max_\theta \sum_{x \in \mathcal{E}_t} \log p_\theta(x), \qquad \theta_{t+1} \leftarrow (1 - \beta)\,\theta_t + \beta\,\hat{\theta}_{t+1}, \tag{5}$$

with smoothing factor $\beta \in (0, 1]$. Because the update depends only on ranks and sampling, it is derivative-free and robust to nonconvexity, which explains its popular use in high-dimensional control and model-based RL planning. In these applications, $x$ denotes an action sequence. We keep our development generic.

**Ensemble CEM.** To mitigate CEM's intrinsic mode-seeking nature, ensemble methods run $K$ CEM instances ("workers") in parallel, each updating according to equation 5

$$\big\{\, \theta_t^{(k)},\; p_{\theta_t^{(k)}} \,\big\}_{k=1}^{K}, \qquad t = 0, 1, \ldots, T.$$

This effectively extends the parametric model to the mixture family that is capable of handling complex, multimodal cost landscapes. We broadly categorize ensemble CEM into two major groups based on the *communication* between workers: centralized methods merge elites across all workers and fit $\{p_{\theta_t^{(k)}}\}$ to mixture models (e.g., Gaussian mixture Okada & Taniguchi (2020)) but add Expectation-Maximization(EM) step overhead and require careful tuning (notably the number of components). Instead, we adopt a more scalable decentralized approach (*DecentCEM* in (Zhang et al., 2022)): workers update independently via equation 5 using disjoint, parallel samples and do not share elites or fit a mixture. The final solution is the best candidate produced by any worker at termination. This "more workers + best-pooling" heuristic is simple to implement and scales well for high-dimensional planning problems.

**Motivation: where decentralization falls short** Despite scalability and parallelism, decentralized CEM remains vulnerable to: **1)** Premature consensus. Independent workers still tend to shrink their covariances and converge to nearby modes under poor initializations. **2)** Independent workers often duplicate search effort, wasting rollouts on already-explored regions. **3)** Pointwise learning. CEM updates use only elite sets, so the ensemble as a whole does not learn the underlying structure; for example, what's each worker's contribution to the ensemble? To address these limitations, we propose a lightweight upgrade of the decentralized approach that aggregates and guides CEM workers in a principled way with minimal overhead.

## 4 (Revised) Method

We start from the decentralized approach in which $n$ independent CEM workers run in parallel with parameters $\{\theta_i\}_{i=1}^{n}$. We drop the time index for brevity and keep a potential $F$ in $D_F$ for general development ($F$ is specialized only in exponential families).

**Bregman Centroid: what represents the ensemble?** To summarize the ensemble effectively, we propose a weighted Bregman centroid of the workers' parameters,

$$\textsc{Bregman Centroid:}\ \boldsymbol{\theta}_c \;=\; \arg\min_{\theta \in \Theta} \sum_{i=1}^{n} w_i\, \mathrm{D}_F\big(\theta_i \,\big\|\, \theta\big), \tag{6}$$

with importance weight $w_i$ that reflects worker $\theta_i$'s solution quality. Intuitively, this centroid $\boldsymbol{\theta}_c$ is a *geometric average* that pulls toward low-cost workers while still reflecting the spread of the ensemble. In Figure 3, the star illustrates this "representative" role.

**Relevance scores: who contributes to the ensemble?** We define a *relevance score* by measuring its Bregman divergence to the centroid, that is, the contribution to the minimized value in equation 6

$$\textsc{Relevance score:}\ \gamma_i \;=\; \mathrm{D}_F(\theta_i \,\|\, \theta_c). \tag{7}$$

Small $\gamma_i$ signals a low-quality worker (small $w_i$ due to poor performance) or a redundant worker (sitting right on the consensus). By ranking workers based on relevance scores, we obtain a compact summary of each worker's unique contribution to the ensemble.

**Trust region: where to respawn new workers?** To keep the ensemble efficient, we prune it by replacing the lowest-scoring workers with new candidates sampled inside a *Bregman ball*:

$$\textsc{Trust Region:}\ \mathcal{B}_\Delta(\theta_c) \;=\; \{\theta : \mathrm{D}_F(\theta\|\theta_c) \leq \Delta\}. \tag{8}$$

New workers start near the ensemble's belief, without collapsing to the centroid due to the diversity threshold $\Delta$. In exponential families, this trust region has Euclidean-like structure, which keeps sampling fast and efficient even in high dimensions (see next sections).

**Algorithm: Bregman–Centroid Guided Evolution Strategy.** With these three components, we present $\mathcal{BC}$-**EvoCEM**, a lightweight upgrade of decentralized CEM ensembles. Each iteration has three stages and leaves every worker's inner CEM step unchanged. The workflow is as follows:

1. **Independent update.** Each of the $n$ workers runs one CEM step with its own population and elites, updating its parameter $\theta_i$ as usual (same as DecentCEM).

2. **Ensemble Summary.** After each update, we assign a weight $w_i$ to each worker based on its cost, compute the *Bregman centroid* $\theta_c$ of the set $\{\theta_i\}_{i=1}^n$, and score each worker by its divergence from the centroid.

3. **Targeted evolution.** Given a radius $\Delta$, we replace the lowest-relevance worker(s) by sampling candidates from a Bregman trust region $\mathcal{B}_\Delta(\theta_c) = \{\theta : \mathrm{D}_F(\theta\|\boldsymbol{\theta}_c) \le \Delta\}$.

The idea of $\mathcal{BC}$-**EvoCEM** is simple (see Algorithm 1): the centroid is the ensemble's consensus of promising regions, the relevance score tells each CEM instance's contribution, and the trust region tells where to add diversity. The loop aims to prevent collapse and use computation budget where it matters most.

---

**Algorithm 1** $\mathcal{BC}$-**EvoCEM** (per iteration)

---

**Require:** Workers $\{\theta_i\}_{i=1}^n$, cost $J(\cdot)$, divergence $D$ (from potential $F := \Psi$), radius $\Delta$
1: **for** $t = 1$ **to** $T$ **do**
2:     **(Workers improve)** $\{\theta_i\} \leftarrow \text{DistributedCEM}(\{\theta_i\}, J)$     ▷ Independent, vanilla per-worker CEM.
3:     **(Weights)** $w_i \leftarrow \text{PerfWeight}(\text{elites/costs of worker } i)$     ▷ Normalize $\sum_i w_i = 1$.
4:     **(Centroid)** $\boldsymbol{\theta}_c \leftarrow \arg\min_\theta \sum_i w_i \mathrm{D}_F(\theta_i\|\theta)$
5:     **(Scores)** $\gamma_i \leftarrow D_F(\theta_i \,\|\, \boldsymbol{\theta}_c)$     ▷ Worker $i$'s contribution to ensemble.
6:     **(Targeted evolution)** $i^\star \leftarrow \arg\min_i \gamma_i$     ▷ Least helpful (redundant/low-quality) worker.
7:         $\theta_{i^\star} \leftarrow \text{TrustRegionSample}(\boldsymbol{\theta}_c, \Delta)$     ▷ Exact (proxy) sampler; see below.
8: **end for**
9: **Return** lowest-cost candidate, and summary $(\boldsymbol{\theta}_c, \{\gamma_i\})$

---

**(main results ONLY) $\mathcal{BC}$-EvoCEM in Exponential Families** We show that three core operations in $\mathcal{BC}$-**EvoCEM**—namely, computing the centroid, scoring and sampling from the trust-region—incur negligible computation overhead in the CEM. Throughout we work with regular, minimal exponential families in their mean coordinate $\{\eta_i\}$ with its corresponding Bregman divergence $\mathrm{D}_{\Psi^*}$ (see Preliminaries). Let $\boldsymbol{\eta}_c$ denote the corresponding Bregman centroid. We summarize the main results here and provide a detailed exposition in Appendix A.

- **Centroid = average of statistics.** With weights $\{w_i\}$ and worker means $\{\eta_i\}$, the centroid satisfies $\boldsymbol{\eta}_c = \sum_i w_i \eta_i$ and $\boldsymbol{\theta}_c = (\nabla\Psi)^{-1}(\boldsymbol{\eta}_c)$. Because CEM already computes the sufficient-statistic averages, the centroid comes essentially for free.

- **Relevance scores = Log-likelihood evaluation.** $\gamma_i = \mathrm{D}_{\Psi^*}(\eta_i\|\boldsymbol{\eta}_c)$ can be evaluated as a log-likelihood proxy against the aggregated statistic $\boldsymbol{\eta}_c$. The computation reduces to one inner product and one cumulant evaluation per worker.

- **Efficient Trust-region sampling.** Drawing new $\eta$ from $\mathcal{B}_\Delta(\boldsymbol{\eta}_c)$ follows a simple, intuitive recipe: pick a random unit direction $u$, start at centroid $\boldsymbol{\eta}_c$, and move along $u$ until it hits the boundary set by $\Delta$. We provide two samplers (see Algorithm 3 and 4 in Appendix): one is **exact** by root-solving the boundary and one **proxy** by quadratically approximating the cumulant.

### 4.1 **(NEW)** Example: $\mathcal{BC}$-**EvoCEM with Gaussians**

We give a concrete example of $\mathcal{BC}$-**EvoCEM**. Consider minimizing a cost $J(x)$ with $n$ CEM instances whose sampling distributions are isotropic Gaussians, $p_{\theta_i} = \mathcal{N}(\mu_i, \sigma^2 I)$ with fixed $\sigma$. For this EF member, the mean parameter $\eta = \mu$ and natural parameter $\theta = \mu/\sigma^2$ (relevant quantities are summarized in Table 1).

Table 1: Closed forms (constants omitted).

| Quantity | Closed form |
|---|---|
| Natural $\leftrightarrow$ Mean | $\theta = \mu/\sigma^2, \quad \eta = \mu$ |
| Cumulant | $\Psi(\theta) = \frac{\sigma^2}{2}\|\theta\|_2^2$ |
| Sufficient statistic | $T(x) = x$ |
| Centroid (mean) | $\mu_c = \eta_c = \sum_i w_i\,\mu_i$ |
| Centroid (natural) | $\theta_c = (\nabla\Psi)^{-1}(\eta_c) = \mu_c/\sigma^2$ |
| Distance score (worker $i$) | $\gamma_i := D_{\Psi^*}(\mu_i\|\mu_c) = \frac{1}{2\sigma^2}\|\mu_i - \mu_c\|_2^2 = \frac{\sigma^2}{2}\|\theta_i - \theta_c\|_2^2$ |
| Local Hessian for sampling | $H = \nabla^2\Psi^*(\eta_c) = I/\sigma^2$ |
| Trust region (mean) | $\{\mu :\ \|\mu - \mu_c\|_2 \le \sqrt{2\Delta}\,\sigma\}$ |
| Trust region (natural) | $\{\theta :\ \|\theta - \theta_c\|_2 \le \sqrt{2\Delta}/\sigma\}$ |

The workflow of $\mathcal{BC}$-**EvoCEM** on this simple EF member is as follows:

- **Step 1: Independent CEM updates.** Each worker runs vanilla CEM on $J(x)$: producing elite set and updating $(\mu_i, \sigma^2)$ (empirical mean of elites in this case). No communication.

- **Step 2 : Performance weights.** Compute $w_i$ from each worker's elite cost and normalize.

- **Step 3 : Centroid.** In mean coordinates: $\mu_c = \eta_c = \sum_{i=1}^n w_i\,\mu_i$. Because $\nabla\Psi$ is linear here, this is also the weighted average in natural coordinates: $\theta_c = \sum_i w_i\theta_i = \mu_c/\sigma^2$.

- **Step 4 : Relevance scores.** For isotropic Gaussians, score becomes squared Euclidean distances

$$\gamma_i = \frac{1}{2\sigma^2}\,\|\mu_i - \mu_c\|_2^2, \quad \text{equivalently,} \quad \frac{\sigma^2}{2}\|\theta_i - \theta_c\|_2^2.$$

- **Step 5 : Trust region & sampling.** Because $\Psi$ is *quadratic*, the trust-region is Euclidean in both mean and natural space. Given the constant Hessian $H = \sigma^{-2}I$, the proxy and exact samplers coincide. Sampling becomes: draw $v \sim \text{Unif}(\mathbb{S}^{d-1})$, $u \sim \text{Unif}[0,1]$, set $r = \sqrt{2\Delta}\,\sigma \cdot u^{1/d}$, then $\mu_{\text{new}} = \mu_c + r\,v$, with $\theta_{\text{new}} = \mu_{\text{new}}/\sigma^2$. Finally, let $i^\star = \arg\min_i \gamma_i$ and replace worker $i^\star$.

Net add-on of $\mathcal{BC}$-**EvoCEM** to vanilla CEM: *(i)* average the means, *(ii)* rank by squared distance to that average, *(iii)* respawn the farthest inside a Euclidean ball. The entire mechanism is a few vector operations:

| Centroid: | $O(nd)$ | weighted mean of existing $\mu_i$ (already in CEM logs). |
|---|---|---|
| Scores: | $O(d)$ per worker | one norm $\|\mu_i - \mu_c\|^2$. |
| Sampling: | $O(d)$ | one random direction + one scalar radius. |

## 5 (Revised) Bregman Centroid–Guided MPC for MBRL

In model-based reinforcement learning (MBRL), we wrap $\mathcal{BC}$-**EvoCEM** into a standard MPC loop. At time $t$, MPC plans an open-loop action sequence $\mathbf{a}_{t:t+H-1} \in \mathcal{A}^H$ under a learned transition model $\tilde{f}$,

$$\max_{\mathbf{a}_{t:t+H-1}} \hat{J}_t(\mathbf{a}_{t:t+H-1}) = \mathbb{E}_{\tilde{f}}\Big[\sum_{\tau=t}^{t+H-1} \gamma^{\tau-t}\,\tilde{r}(s_\tau, a_\tau)\Big] \quad \text{s.t.}\ \ s_{\tau+1} \sim \tilde{f}(s_\tau, a_\tau),$$

executes only the first action, and replans at $t+1$. We keep $K$ CEM workers fully *independent* internally, but couple them between MPC steps via a solution-weighted Bregman centroid used to warm start all workers.

Concretely, after each solve: (i) compute the centroid over worker distributions; (ii) initialize all workers at $t+1$ from this centroid instead of shifting a single solution (Chua et al., 2018; Zhang et al., 2022); and (iii) to prevent ensemble collapse, periodically *respawn* the lowest-relevance workers by sampling within

a centroid-centered Bregman trust region. This preserves every worker's inner CEM loop (for minimal overhead), adds only a few vector operations per control step, and balances consensus with targeted diversity, effectively serving as a drop-in, plug-and-play MPC wrapper for any planning-based MBRL baselines (see Algorithm 2).

---

**Algorithm 2** Drop-in MPC Wrapper for MBRL. Add-ons operations are highlighted by **Orange**.

---

**Require:** $K$ CEM workers, buffer $\mathcal{D}$, horizon $H$, inner iterations $L$, respawn frequency $P$

1: **for** each training iteration **do**
2:      // **Training dynamics model on data buffer.** //
3:      Train dynamics model $\tilde{f}$ on $\mathcal{D}$
4:      Initialize worker params $\{\theta_k\}_{k=1}^K$
5:      // **Running MPC policy.** //
6:      **for** each control step $t = 1, \ldots, H$ **do**
7:          // **Warm-start CEMs by the centroid.** //
8:          Compute Bregman centroid $\boldsymbol{\theta}_c \leftarrow \text{CENTROID}(\{\theta_k\}; \{w_k\})$
9:          Warm-start: $\theta_k^{(0)} \leftarrow \boldsymbol{\theta}_c$
10:         // **Trajectory Optimization by CEM.** //                ▷ CEM internal unchanged
11:         **CEM**: For each $k$, run $L$ CEM iterations from $\theta^{(0)}$ under $\tilde{f}$; obtain $\theta_k \leftarrow \theta_k^{(L)}$ and $\hat{J}_k^{\text{best}}$
12:         // **Evolution strategy.** //
13:         **Score & Replace**: if $t \bmod P = 0$ then
14:           Replace worst workers via the *Target Evolution* step in 1
15:         Execute the 1st action of $\arg\max_k \hat{J}_k^{\text{best}}$
16:         Append new transition to $\mathcal{D}$ and advance the environment
17:      **end for**
18: **end for**

---

# 6  (NEW) Discussion: Why Bregman divergence and its centroid?

In this section, we justify the use of the Bregman divergence and its centroid in the CEM. We adopt the notations introduced in the Preliminaries and repeatedly invoke the fact that CEM is a KL-projection of the sampling distribution (elites' law) onto the parametric family $\{p_\theta\}$ (standard result of CEM as KL minimization, see (Rubinstein & Kroese, 2004)).

**Ensemble CEM as a KL–projection.** In an ensemble CEM, $n$ workers maintain EF members $\{p_{\theta_i}\}_{i=1}^n$ and produce nonnegative weights $w_i$ with $\sum_i w_i = 1$ from their elite sets. A natural ensemble summary is to approximate the mixture

$$p_{\text{mix}} := \sum_{i=1}^n w_i\, p_{\theta_i}$$

by a *single* distribution $p_\theta$ within the same parametric family. Using the "CEM as a KL-projection" fact, one obtains the following.

**Proposition 1** (Ensemble CEM as KL-projection). *Let $p_{mix} = \sum_i w_i p_{\theta_i}$. Then the three objectives*

$$\boldsymbol{\theta}_c = \arg\min_\theta \sum_{i=1}^n w_i \, \text{KL}\big(p_{\theta_i} \,\|\, p_\theta\big) = \arg\min_\theta \text{KL}\big(p_{mix} \,\|\, p_\theta\big) = \arg\max_\theta \mathbb{E}_{p_{mix}}\big[\log p_\theta(X)\big]$$

*are equivalent and share the same minimizer $\boldsymbol{\theta}_c$.*

Specializing to exponential families yields the following results:

**Corollary 1** (EF: moment matching & centroid). *If $p_\theta$ is a regular, minimal EF with cumulant $\Psi$, then*

$$\mathbb{E}_{p_{mix}}[\log p_\theta(X)] = \big\langle \theta,\ \mathbb{E}_{p_{mix}}[T(X)] \big\rangle - \Psi(\theta) + const,$$

*so the maximizer $\boldsymbol{\theta}_c$ of Prop. 1 satisfies the moment-matching condition*

$$\nabla\Psi(\boldsymbol{\theta}_c) = \mathbb{E}_{p_{mix}}[T(X)] = \sum_{i=1}^{n} w_i\,\eta_i.$$

*Equivalently, $\boldsymbol{\theta}_c$ is the* left-sided *Bregman centroid in natural space, and, dually, $\boldsymbol{\eta}_c = \sum_i w_i\eta_i$ is the* right-sided *centroid in mean space with generator $\Psi^*$.*

These results show two crucial facts: (i) the aggregate stays inside the family (**in-family closure property**), and (ii) the ensemble's best single EF representative is the member whose sufficient-statistic mean matches that of the mixture (**KL-projection onto the family**). No extra heuristic is introduced: the choice of Bregman divergence and its centroid is a direct consequence of EF geometry and KL alignment.

**Other alternative distances/divergences.** Below we summarize several commonly used notions of a *central* distribution within a collection and discuss why each is inadequate in the CEM setting.

- **Euclidean/Mahalanobis in parameter space.** Sensitive to reparameterization; does not align with log-likelihood; no moment-matching guarantee; may push updates outside EF geometry.

- **Kernel/Maximum mean discrepancy(MMD) centroids.** Require careful feature selection and large Gram matrices; the induced centroid generally lies outside the EF and lacks a simple moment-matching form; computationally heavier than the closed-form EF centroid.

- **Wasserstein barycenters.** Geometry-preserving, powerful but iterative, high-dimensional costs (e.g., Sinkhorn solve), and no EF closure in general; the objective is not the one CEM already optimizes.

- **Other $f$-divergences.** No EF-KL identity in general; lose the clean moment-matching and the right-sided likelihood interpretation.

For EF aggregation inside CEM, KL (namely, Bregman with generator $\Psi^*$) is unique in giving (i) in-family closure, (ii) a closed-form, moment-matching centroid, and (iii) a principled trust region tied to likelihood via $D_{\Psi^*}$.

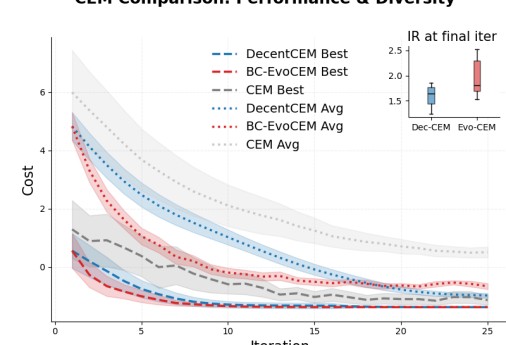

# 7 Experimental Results

## 7.1 Multi-modal optimization

We first demonstrate our method on a multi-modal optimization problem with the cost function (Fig. 1 shows the cost landscape with multiple attraction basins):

$$J(\boldsymbol{x}) = \sin(3x_1) + \cos(3x_2) + 0.5\,\|\boldsymbol{x}\|_2^2.$$

We compare our method against (1) vanilla CEM and (2) decentralized CEM(DecentCEM) (Zhang et al., 2022), with the same parametric distribution $p_\theta = \mathcal{N}(\theta, 0.5^2 I)$.

Figure 3: Performance comparison for vanilla, decentralized, and our CEM methods. Solid/dashed lines show the mean/best cost, shaded bands $\pm 1$ std. Information radius (sample variance in this case) at iter 25 is shown.

Our approach (red in Fig. 3) demonstrates faster convergence in both *Best* and *Average* costs. Importantly, the trust-region sampling maintains solution diversity, as shown by the final IR values (i.e., sample variance in this case).

**Better worker efficiency.** Our scaling study (see Fig.4) reveals a clear $\log_2(\#\text{WORKERS})$ law: beyond 8–16 workers, DecentCEM shows sharp diminishing returns. $\mathcal{BC}$**-EvoCEM** achieves the same best-cost and diversity asymptotics with roughly *half* of the workers. It shows markedly higher ***worker efficiency*** while avoiding the quadratic rollout cost that comes with an increasingly larger ensemble.

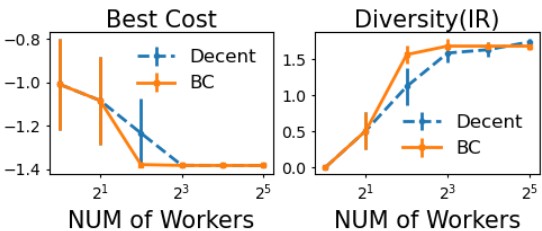

Figure 4: Scaling with workers (20 SEEDS). BC-EvoCEM needs fewer workers to hit the same best-cost.

## 7.2 Navigation Task

We consider a cluttered 2D navigation task with first-order dynamics and time-step $\Delta t = 0.2$. A planning horizon of $H = 200$ yields a $2H$-dimensional action sequence. We employ 5 independent diagonal-Gaussian CEM workers with identical CEM hyper-parameters and initialization. To sample from the trust region in this high-dimensional space, we use the PROXYSAMPLER (see Alg. 4).

Table 2: Normalized costs and relative drop versus decentralized CEM.

| Method | Average cost | | Best cost | |
|---|---|---|---|---|
| | Norm. | Drop (%) | Norm. | Drop (%) |
| Decentralized CEM | 1.00 | — | 1.00 | — |
| BC-evoCEM | 0.18 | 82.4 | 0.55 | 45.3 |

Figure 5 (left) visualizes trajectories from a fully decentralized CEM, which disperse widely and frequently deviate from the start–goal line. In contrast, $\mathcal{BC}$**-EvoCEM** (right) maintains a tight cluster of trajectories around the Bregman-centroid path (green dashed line), producing a more diverse and goal-directed planning. Notably, the centroid itself is not guaranteed to avoid obstacles as it serves only as an information-geometric summary of all workers. Quantitatively, $\mathcal{BC}$**-EvoCEM** yields significant improvements in both average and best cost without incurring noticeable computational overhead.

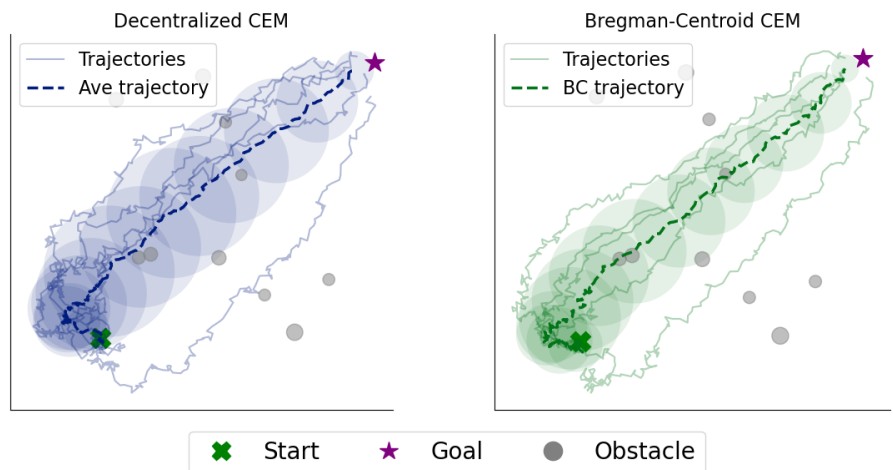

Figure 5: Trajectory distributions from decentralized CEM (left) and Bregman–centroid guided CEM (right) on a point-mass navigation task. The *representatives* of each method (dashed line) are the average and Bregman-centroid trajectory.

### 7.3 Bregman Centroid Guided MPC in MBRL

**Baselines and minimal implementation.** Our study is built on the PETS framework (Chua et al., 2018) and the DECENTCEM implementation of Zhang et al. (2022). Dynamics learning, experience replay, and per-worker CEM updates are left untouched. We wrap the optimizer with a *Bregman-Centroid warm-start* (see the MPC in 2). The wrapper is drop-in and lightweight, i.e. $< 50$ lines of Python code. This plug-and-play feature makes the method readily portable to any planning-based MBRL codebase.

**Protocol.** To isolate the impact of the trajectory optimizer, we keep the rest of PETS fixed and compare three variants: vanilla CEM, DecentCEM, and $\mathcal{BC}$-**EvoCEM** under two dynamics models: 1) a **deterministic** model trained by minimizing MSE, and 2) a **probabilistic ensemble** with trajectory sampling (Chua et al., 2018). Full results and hyper-parameters are in Appendix D.

**Experimental results.** With no additional stochasticity in the pipeline, all performance differences stem from the optimizer itself. BC-evoCEM delivers **faster early learning** and **higher asymptotic return** on four of six benchmarks (see Figure 6). Its sample-efficiency advantage is quantified in Table 3, where $\mathcal{BC}$-**EvoCEM**M reaches 90% of its own final reward up to **4**× sooner than baselines. Vanilla CEM's covariance collapses quickly, and DECENTCEM suffers from independent worker collapse. By contrast, the Bregman centroid keeps workers in promising regions *while preserving ensemble diversity*, translating into sustained exploration and higher returns.

Table 3: Sample–efficiency *boost* (↑) of $\mathcal{BC}$-**EvoCEM** over two CEM variants. Values $> 1$ mean BC-evoCEM learns faster

| Environment | vs. DecentCEM | vs. vanilla CEM |
|---|---|---|
| Acrobot | ×2.1 | ×2.6 |
| CartPole | ×3.4 | ×4.0 |
| Hopper | ×1.0 | ×1.1 |
| Inverted Pendulum | ×1.1 | ×1.2 |
| Reacher | ×1.0 | ×1.1 |
| Pusher | ×1.9 | ×2.0 |

When both *epistemic* and *aleatoric* uncertainty are already captured via bootstrap sampling, the performance differences among the three optimizers become statistically indistinguishable (see Appendix D.3). Here, the model's intrinsic stochasticity supplies sufficient trajectory dispersion, so further optimizer-level exploration yields diminishing returns.

**Uncertainties in MBRL.** The controlled study highlights two distinct yet coupled sources of uncertainty in model-based RL: *model uncertainty* and *optimality uncertainty*. Improving the dynamics model (e.g., probabilistic ensembles) addresses the former, whereas a diversity-informed optimizer (e.g., $\mathcal{BC}$-**EvoCEM**) directly addresses the latter. Once the dynamics model approaches its performance cap (or its representational capacity is bottlenecked), optimality uncertainty predominates; in this regime, geometry-informed exploration such as $\mathcal{BC}$-**EvoCEM** in the action space delivers a complementary boost.

### 7.4 Real-World Quadruped Navigation Experiments

To demonstrate that $\mathcal{BC}$-**EvoCEM** extends beyond simulation, we deploy it on a UNITREE GO2 quadruped tasked with navigating a $4 \times 4$ m cluttered arena (Fig. 7a). Both $\mathcal{BC}$-**EvoCEM** and a vanilla CEM baseline optimize a velocity policy control $= [v_{\text{linear}}, v_{\text{angular}}]$ at $20\,Hz$. Success is defined as reaching the goal without collision within a given number of iterations. We report five metrics in Table 4.

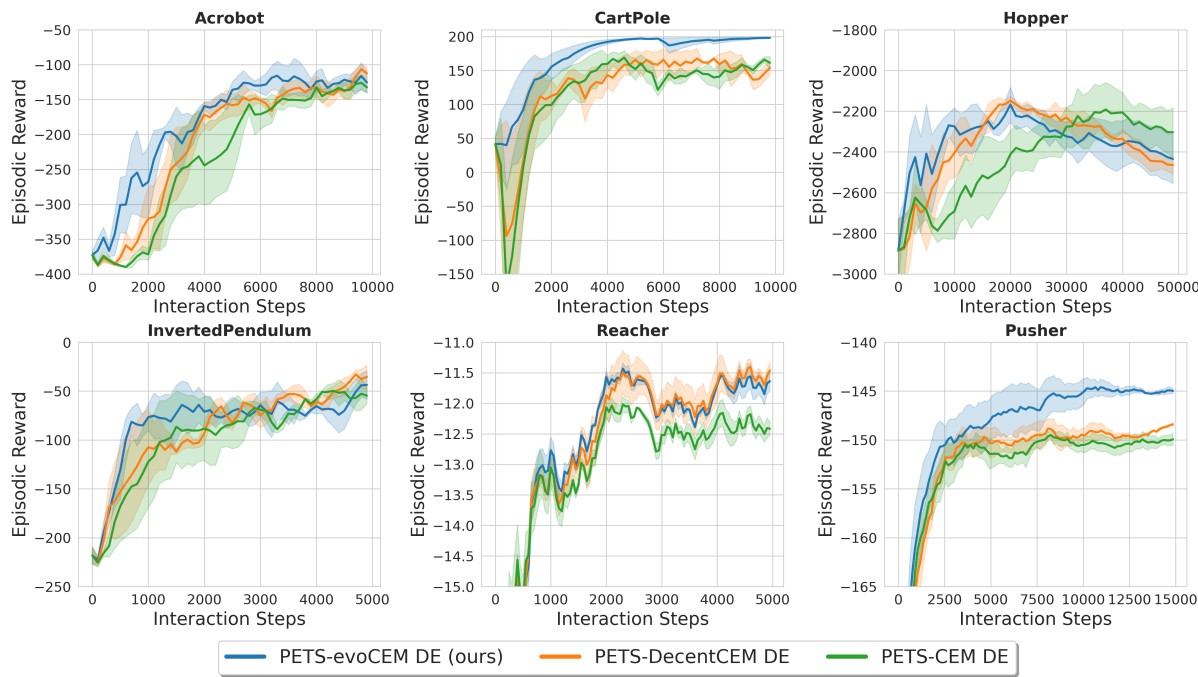

Figure 6: Training return curves across six control tasks using PETS with different CEM-based optimizers. All methods use the deterministic dynamics model. Curves show mean performance over 3 random seeds.

Table 4: Performance metrics (lower ↓ is better).

|  | Success | Eff.↓ | Smooth↓ | Cost↓ | $t_{\text{opt}}$↓ |
|---|---|---|---|---|---|
| BC-evoCEM | ✓ | 1.158 | **10.90** | **21.16** | 0.462 |
| CEM | ✗ | **1.079** | 12.92 | 36.96 | **0.400** |

**Experimental results.** Figure 7 and Table 4 demonstrate that:

- **Planned Path** (Fig. 7). The vanilla CEM rollout drifts into the obstacle cluster and halts ∼0.9 m short of the goal (white circle), whereas $\mathcal{BC}$-**EvoCEM** generates a collision-free path that reaches the goal (yellow star). Its sample cloud (gray) remains tighter and farther from danger zones.

- **Quantitative metrics** (Table 4). Compared to vanilla CEM, $\mathcal{BC}$-**EvoCEM** yields a **smoother** control profile and cuts cumulative cost by **43%**, with only a modest +0.062 s optimization overhead.

These results validate our central claim: $\mathcal{BC}$-**EvoCEM** is a lightweight, drop-in upgrade to existing CEM implementations that delivers substantially safer, smoother, and cheaper trajectories without sacrificing real-time performance.

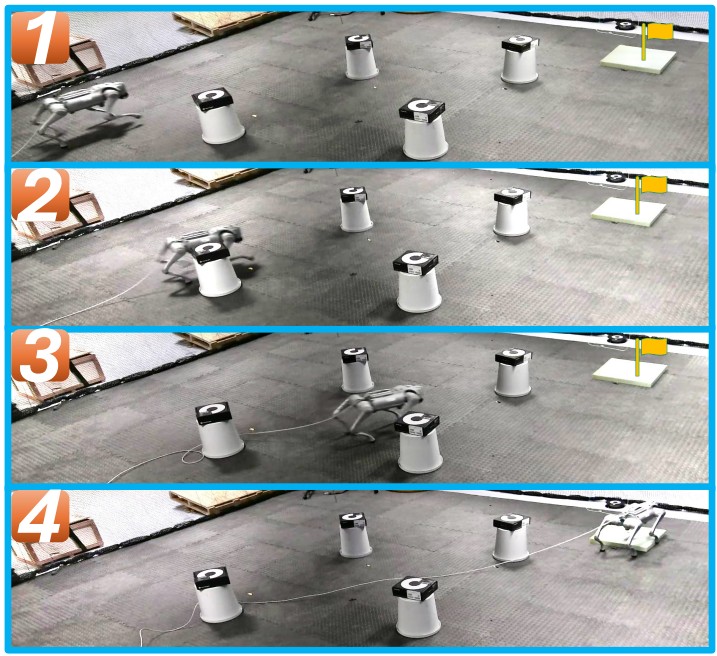

(a) UNITREE GO2 executing BC-evoCEM's plans.

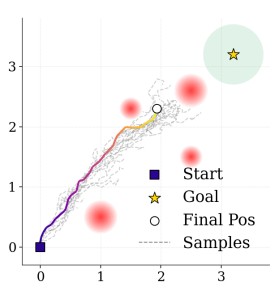

(b) Planned path — CEM.

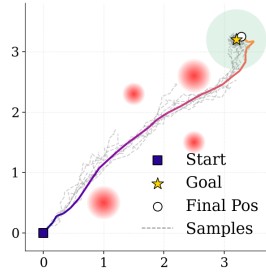

(c) Planned path — BC-evoCEM.

Figure 7: Real-world navigation experiment. Top: sequential onboard snapshots. Bottom: top-down planned paths from each optimizer. Quantitative metrics are reported separately in Table 4.

## 8 Conclusion

We introduced $\mathcal{BC}$**-EvoCEM**, a lightweight ensemble extension of the Cross-Entropy Method that has (i) principled information aggregation and (ii) diversity-driven exploration with near-zero computation overhead. Across optimization problems and model-based RL benchmarks, $\mathcal{BC}$**-EvoCEM** demonstrates faster convergence and attains higher-quality solutions than vanilla and decentralized CEM. Its plug-and-play design enables easy integration into MPC loops while preserving the algorithmic simplicity that makes CEM appealing in the first place.

## (Extended) Limitations

Here we discuss limitations of the proposed $\mathcal{BC}$**-EvoCEM** and provide potential directions for addressing them in future work.

The Bregman trust region is convex. While it improves *local* exploration, it cannot by itself bridge distant CEM solutions. As a result, we pair it with decentralization and restarts. A natural extension is a *multi-center* variant: cluster elites and apply per-cluster Bregman balls (a mixture of trust regions), enabling mode-wise refinement without breaking the lightweight integration.

In addition, all information-geometric arguments (closed-form centroid, ellipsoidal trust region, likelihood-based ranking) hold only for *regular, minimal exponential family* distributions in mean coordinates. This restriction limits the expressiveness of the CEM distributions. Future work will transfer these ideas to richer models via *geometric-preserving* transport maps (Villani et al., 2008). In addition, while we prove the centroid and repawned CEM workers remain inside a Bregman ball, the method still lacks global optimality guarantees and convergence analysis. It inherits these limitations from CEM. A promising direction is to consider the proposed $\mathcal{BC}$**-EvoCEM** in the stochastic mirror-descent framework (Ahn & Chewi, 2021), which potentially provides non-asymptotic convergence bounds via the primal-dual relationship.

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

# A (Extended) BC-evoCEM in Exponential Families

## A.1 Preliminaries

We follow the notation in the main text. Let $\{p_\theta : \theta \in \Theta \subset \mathbb{R}^d\}$ be a regular, minimal exponential family

$$p_\theta(x) = h(x) \exp\{\langle \theta, T(x) \rangle - \Psi(\theta)\},$$

with strictly convex (Legendre-type) cumulant $\Psi$. The mean (expectation) parameter is

$$\eta := \nabla\Psi(\theta) = \mathbb{E}_\theta[T(X)].$$

Let $\Psi^*$ be the convex conjugate of $\Psi$. The associated Bregman divergences are

$$\mathrm{D}_{\Psi^*}(\eta \,\|\, \eta') = \Psi^*(\eta) - \Psi^*(\eta') - \langle \nabla\Psi^*(\eta'),\, \eta - \eta' \rangle, \qquad \mathrm{D}_\Psi(\theta' \,\|\, \theta) = \Psi(\theta') - \Psi(\theta) - \langle \nabla\Psi(\theta),\, \theta' - \theta \rangle.$$

Under the Legendre duality $\eta = \nabla\Psi(\theta)$ and $\theta = \nabla\Psi^*(\eta)$,

$$\mathrm{D}_{\Psi^*}(\eta \,\|\, \eta') \;=\; \mathrm{D}_\Psi(\theta' \,\|\, \theta) \quad \text{with} \quad \eta = \nabla\Psi(\theta),\ \eta' = \nabla\Psi(\theta'). \tag{9}$$

Moreover, for exponential families, we have the following identity with Kullback–Leibler(KL) divergence (see (Amari, 1995))

$$\mathrm{KL}(p_\theta \,\|\, p_{\theta'}) \;=\; D_\Psi(\theta' \,\|\, \theta) \;=\; D_{\Psi^*}(\eta \,\|\, \eta'). \tag{10}$$

**Remark 1** (Notes on the "Sidedness"). *Bregman divergences are **asymmetric**: in general $D_F(x \,\|\, y) \neq D_F(y \,\|\, x)$. Consequently, there are two notion of Bregman centroids for points $\{x_i\}$ with weights $\{w_i\}$:*

$$\textbf{\textit{Right--sided:}} \quad \arg\min_c \ \sum_i w_i \, \mathrm{D}_F(x_i \,\|\, c), \qquad \textit{solution } c = \sum_i w_i \, x_i;$$

$$\textbf{\textit{Left--sided:}} \quad \arg\min_c \ \sum_i w_i \, \mathrm{D}_F(c \,\|\, x_i), \qquad \textit{solution } \nabla F(c) = \sum_i w_i \, \nabla F(x_i),$$

*whenever $F$ is Legendre-type function and $\nabla F$ is invertible (Nielsen & Nock, 2009). In this work we aggregate EF members using the **right--sided** centroid in mean ($\eta$) coordinates with generator $F = \Psi^*$. By equation 9, this is equivalent to the **left--sided** centroid in natural ($\theta$) coordinates with generator $F = \Psi$. This convention matches the EF identity in equation 10: minimizing $\sum_i w_i \, \mathrm{KL}(p_{\theta_i} \,\|\, p_\theta)$ is exactly the left--sided Bregman centroid in $\theta$ and, dually, the right--sided centroid in $\eta$.*

## A.2 Centroid in natural and mean coordinates

As the core of our method, the Bregman centroid admits a simple closed form in mean coordinates.

**Proposition 2** (Bregman centroid). *Given weights $\mathbf{w} = (w_1, \ldots, w_n)$ with $w_i \geq 0$ and $\sum_i w_i = 1$, and EF members with parameters $(\theta_i, \eta_i)$ related by $\eta_i = \nabla\Psi(\theta_i)$, the following are equivalent and admit a unique solution:*

$$\boldsymbol{\eta}_c = \arg\min_\eta \ \sum_{i=1}^n w_i \, \mathrm{D}_{\Psi^*}(\eta_i \,\|\, \eta) \qquad \Longleftrightarrow \qquad \boldsymbol{\eta}_c = \sum_{i=1}^n w_i \, \eta_i, \tag{11}$$

$$\boldsymbol{\theta}_c = \arg\min_\theta \ \sum_{i=1}^n w_i \, \mathrm{D}_\Psi(\theta \,\|\, \theta_i) \qquad \Longleftrightarrow \qquad \nabla\Psi(\boldsymbol{\theta}_c) = \sum_{i=1}^n w_i \, \eta_i. \tag{12}$$

*Equivalently, $\boldsymbol{\theta}_c = (\nabla\Psi)^{-1}(\boldsymbol{\eta}_c)$.*

*Proof.* The right–sided identity equation 11 is the *mean-as-minimizer* property of Bregman divergences with generator $\Psi^*$ (Banerjee et al., 2005). The equivalence with equation 12 follows from the duality in equation 9 and the invertibility of $\nabla\Psi$ for Legendre-type function $\Psi$ (Nielsen & Nock, 2009; Rockafellar, 2015). □

CEM's likelihood evaluations (see equation 1) naturally yield empirical sufficient-statistic means for each member,

$$\widehat{\eta}_i \;=\; \frac{1}{N} \sum_{j=1}^{N} T(x_{i,j}),$$

so the centroid in mean space is obtained *for free* as $\boldsymbol{\eta}_c = \sum_{i=1}^{n} w_i \widehat{\eta}_i$, and the corresponding natural parameter is $\boldsymbol{\theta}_c = (\nabla \Psi)^{-1}(\boldsymbol{\eta}_c)$, with no extra optimization (e.g., solving a separate weighted-centroid problem).

### A.3 Relevance score: log-likelihood evaluation

**Log-likelihood and Bregman Divergence.** In this subsection, we ignore the base measure $h(x)$ in $p_\theta(x)$, since it does not depend on $\theta$. Write $t$ for as a sufficient-statistic value or an empirical mean (e.g., $\widehat{\eta}$). The parametric log-likelihood of an EF member then given by

$$\ell(\theta; t) = \langle \theta, t \rangle - \Psi(\theta). \tag{13}$$

Using $\Psi^*(\eta) = \langle \eta, \theta \rangle - \Psi(\theta)$ and the definition of $D_{\Psi^*}$, we have

$$
\begin{aligned}
\ell(\theta; t) &= \langle \theta, t \rangle - \Psi(\theta) \\
&= \Psi^*(t) - \Big( \Psi^*(t) - \langle \theta, t \rangle + \Psi(\theta) \Big) \\
&= \Psi^*(t) \;-\; D_{\Psi^*}\big(t \parallel \eta\big).
\end{aligned}
\tag{14}
$$

Thus, up to the constant $\Psi^*(t)$, maximizing log-likelihood is equivalent to *minimizing the Bregman divergence* $D_{\Psi^*}(t \| \eta)$.

**Relevance score.** Let $\boldsymbol{\eta}_c = \sum_i w_i \widehat{\eta}_i$ be the centroid in the mean space and $\boldsymbol{\theta}_c = \nabla \Psi^*(\boldsymbol{\eta}_c)$ its natural parameter. For a worker $i$ with empirical mean $\widehat{\eta}_i$, evaluate the centroid at $t = \widehat{\eta}_i$:

$$\ell(\boldsymbol{\theta}_c; \widehat{\eta}_i) = \Psi^*(\widehat{\eta}_i) - D_{\Psi^*}\big(\widehat{\eta}_i \parallel \boldsymbol{\eta}_c\big). \tag{15}$$

If $\theta_i^\star$ denotes the worker-specific MLE (so $\nabla \Psi(\theta_i^\star) = \widehat{\eta}_i$), then

$$\underbrace{\ell(\boldsymbol{\theta}_c; \widehat{\eta}_i) - \ell(\theta_i^\star; \widehat{\eta}_i)}_{\text{log-likelihood drop of centroid on worker } i} \;=\; - D_{\Psi^*}\big(\widehat{\eta}_i \parallel \boldsymbol{\eta}_c\big) \tag{16}$$

This motivates our definition of *relevance score* of worker $i$ with respect to the centroid:

$$\gamma_i \;:=\; -\big(\ell(\boldsymbol{\theta}_c; \widehat{\eta}_i) - \ell(\theta_i^\star; \widehat{\eta}_i)\big) \;=\; D_{\Psi^*}\big(\widehat{\eta}_i \parallel \boldsymbol{\eta}_c\big), \tag{17}$$

which measures the negative log-likelihood(NLL) gap as a discrepancy score.

**Practical considerations.** In most cases, we work with an EF member (e.g., Gaussians) with a known conjugate $\Psi^*$. Even when $\Psi^*$ is not available in closed form, equation 15 and equation 17 can be computed using only the cumulant $\Psi$ and $(\boldsymbol{\theta}_c, \boldsymbol{\eta}_c)$:

$$
\begin{aligned}
D_{\Psi^*}\big(\widehat{\eta}_i \parallel \boldsymbol{\eta}_c\big) &= \Psi^*(\widehat{\eta}_i) - \Psi^*(\boldsymbol{\eta}_c) - \big\langle \nabla \Psi^*(\boldsymbol{\eta}_c), \, \widehat{\eta}_i - \boldsymbol{\eta}_c \big\rangle \\
&= \Psi^*(\widehat{\eta}_i) \;-\; \big\langle \boldsymbol{\theta}_c, \widehat{\eta}_i \big\rangle \;+\; \Psi(\boldsymbol{\theta}_c),
\end{aligned}
\tag{18}
$$

since $\nabla \Psi^*(\boldsymbol{\eta}_c) = \boldsymbol{\theta}_c$ and $\Psi^*(\boldsymbol{\eta}_c) = \langle \boldsymbol{\eta}_c, \boldsymbol{\theta}_c \rangle - \Psi(\boldsymbol{\theta}_c)$. Consequently,

$$\ell(\boldsymbol{\theta}_c; \widehat{\eta}_i) = \big\langle \boldsymbol{\theta}_c, \, \widehat{\eta}_i \big\rangle - \Psi(\boldsymbol{\theta}_c) \;=\; \Psi^*(\widehat{\eta}_i) - D_{\Psi^*}\big(\widehat{\eta}_i \parallel \boldsymbol{\eta}_c\big),$$

and the relevance score equation 17 follows immediately.

### A.4 Trust–region sampling

We sample inside the Bregman ball trust-region in mean space,

$$\mathcal{S}_\Delta(\eta_c) := \big\{\eta \in \mathcal{E} : \ \mathrm{D}_{\Psi^*}(\eta \,\|\, \boldsymbol{\eta}_c) \le \Delta\big\},$$

which is star-shaped about $\boldsymbol{\eta}_c$ along every Euclidean ray (see Lemma 1 below). We follow the standard ray-shooting recipe for sampling in convex bodies (Schneider, 2013): draw a random direction $v$, find the boundary distance $\rho_{\max}(v)$ along $v$, then sample the radius with the appropriate volume correction.

---

**Algorithm 3** TRUSTREGIONSAMPLER (Exact)

---

**Require:** Centroid $\eta_c$, radius $\Delta$, Bregman divergence $D_{\Psi^*}$        $\triangleright$ $\Psi^*$ is the convex conjugate.
1: Draw direction $v \sim \mathrm{Unif}(\mathbb{S}^{d-1})$        $\triangleright$ Random ray from the unit sphere.
2: Find $\rho_{\max}$ s.t. $D_{\Psi^*}(\boldsymbol{\eta}_c + \rho_{\max}v \,\|\, \boldsymbol{\eta}_c) = \Delta$        $\triangleright$ 1D root solve for the boundary.
3: Sample $u \sim \mathrm{Unif}[0,1]$; set $\eta_{\mathrm{new}} \leftarrow \eta_c + u^{1/d}\,\rho_{\max}\,v$        $\triangleright$ Volume correction for uniform law.
4: **return** $\eta_{\mathrm{new}}$, optional $\theta_{\mathrm{new}} \leftarrow (\nabla\Psi)^{-1}(\eta_{\mathrm{new}})$

---

**Theorem 1.** *Let $\rho_{\max}(v)$ be the unique solution of $\mathrm{D}_{\Psi^*}(\boldsymbol{\eta}_c + \rho v \,\|\, \boldsymbol{\eta}_c) = \Delta$. Algorithm 3 returns $\eta_{\mathrm{new}} \sim \mathrm{Unif}\big(\mathcal{S}_\Delta(\eta_c)\big)$. Under the dual map $\theta = \nabla\Psi^*(\eta)$, the image set is the left-sided natural-parameter ball*

$$\mathcal{B}_\Delta(\boldsymbol{\theta}_c) := \big\{\theta \in \Theta : \ \mathrm{D}_\Psi(\boldsymbol{\theta}_c \,\|\, \theta) \le \Delta\big\},$$

*and $\theta_{\mathrm{new}} \in \mathcal{B}_\Delta(\boldsymbol{\theta}_c)$. If $\Psi$ is quadratic (so $\nabla^2\Psi$ is constant), then $\theta_{\mathrm{new}}$ is uniform on $\mathcal{B}_\Delta(\boldsymbol{\theta}_c)$ with respect to Lebesgue measure.*

---

**Algorithm 4** TRUSTREGIONSAMPLER (Proxy; ellipsoidal approximation)

---

**Require:** Centroid $\boldsymbol{\eta}_c$, radius $\Delta$, $H \leftarrow \nabla^2\Psi^*(\boldsymbol{\eta}_c)$        $\triangleright$ Local metric (cheap and closed-form).
1: Draw $v \sim \mathrm{Unif}(\mathbb{S}^{d-1})$
2: $\widehat{\rho}_{\max} \leftarrow \sqrt{2\Delta/(v^\top Hv)}$        $\triangleright$ Boundary along direction $v$.
3: Sample $u \sim \mathrm{Unif}[0,1]$; set $\eta_{\mathrm{new}} \leftarrow \eta_c + u^{1/d}\,\widehat{\rho}_{\max}\,v$        $\triangleright$ Volume correction for uniform law.
4: **return** $\eta_{\mathrm{new}}$, optional $\theta_{\mathrm{new}} \leftarrow (\nabla\Psi)^{-1}(\eta_{\mathrm{new}})$

---

**Proxy Sampler.** To avoid per–direction root solving in Algorithm 3, we locally approximate the trust region

$$\mathcal{S}_\Delta(\boldsymbol{\eta}_c) := \big\{\eta \in \mathcal{E} : \mathrm{D}_{\Psi^*}(\eta \,\|\, \boldsymbol{\eta}_c) \le \Delta\big\}.$$

Fix the direction $v \in \mathbb{S}^{d-1}$ and define the radial function

$$g_v(\rho) := \mathrm{D}_{\Psi^*}(\boldsymbol{\eta}_c + \rho v \,\|\, \boldsymbol{\eta}_c), \qquad \rho \ge 0.$$

Then $g_v(0) = 0$ and $g'_v(0) = \langle \nabla\Psi^*(\boldsymbol{\eta}_c) - \nabla\Psi^*(\boldsymbol{\eta}_c), v\rangle = 0$. A Taylor expansion about $\rho = 0$ gives

$$g_v(\rho) \ = \ \frac{1}{2}\rho^2\, v^\top \underbrace{\nabla^2\Psi^*(\boldsymbol{\eta}_c)}_{=:\,H}\, v \ + \ \mathcal{O}(\rho^3) \ \approx \ \frac{1}{2}\rho^2\, v^\top H\, v, \tag{19}$$

with

$$H = \nabla^2\Psi^*(\boldsymbol{\eta}_c) = \big[\nabla^2\Psi(\boldsymbol{\theta}_c)\big]^{-1}.$$

Substituting equation 19 into the trust-region constraint $g_v(\rho) \le \Delta$ yields the directional bound

$$\frac{1}{2}\rho^2\, v^\top H\, v \ \le \ \Delta \quad \implies \quad \rho \ \le \ \widehat{\rho}_{\max}(v) := \sqrt{\frac{2\Delta}{v^\top H\, v}}.$$

Hence the proxy trust region in mean space is the Mahalanobis (ellipsoidal) ball

$$\widehat{\mathcal{S}} = \big\{\eta \in \mathcal{E} : \ (\eta - \boldsymbol{\eta}_c)^\top H\, (\eta - \boldsymbol{\eta}_c) \le 2\Delta\big\}.$$

**Diagonal Gaussian case (fixed variances).** For $p(x) = \mathcal{N}(\mu, \operatorname{diag}(\sigma^2))$ with *fixed* diagonal variances, the natural parameter is $\theta = \operatorname{diag}(\sigma^{-2})\,\mu$ and

$$\Psi(\theta) = \tfrac{1}{2}\sum_{i=1}^{d}\sigma_i^2\theta_i^2 + \text{const}, \qquad \eta = \nabla\Psi(\theta) = \mu, \qquad \Psi^*(\eta) = \tfrac{1}{2}\sum_{i=1}^{d}\frac{\eta_i^2}{\sigma_i^2} + \text{const}.$$

Therefore

$$H = \nabla^2\Psi^*(\eta) = \operatorname{diag}\!\big(\sigma_1^{-2}, \ldots, \sigma_d^{-2}\big),$$

and the proxy ball $\widehat{\mathcal{S}}$ is *axis-aligned*:

$$\widehat{\mathcal{S}} = \Big\{\eta : \ \sum_{i=1}^{d}\frac{(\eta_i - (\boldsymbol{\eta}_c)_i)^2}{\sigma_i^2} \le 2\Delta\Big\}.$$

In particular, along any coordinate axis $e_i$ the directional radius is $\widehat{\rho}_{\max}(e_i) = \sqrt{2\Delta}\,\sigma_i$, so the extremal coordinates are $(\boldsymbol{\eta}_c)_i \pm \sqrt{2\Delta}\,\sigma_i$.

**Practical considerations.** During CEM update of full diagonal Gaussians, the empirical covariance often collapses, becoming low–rank or even singular; in other words, the *mean* component $\mu$ quickly dominates the search directions. A practical trick here is to *freeze* the diagonal variance vector $\sigma^2$ after a few iterations (or to enforce a fixed lower bound). In practice, we perform such fix-variance trick during the trust region sampling step for high-dimensional planning tasks, including the MPC implementation for MBRL in Sec. 7.3. Because the Hessian matrix is block–diagonal (a diagonal sub-block for the mean and a sub-block for the variances), we can safely use the approximation to perform such *coordinate-wise* updates **exclusively** on the mean block. This also avoids numerical issues from near-singular covariances.

# B (NEW) BC-evoCEM in MBRL

## B.1 Background: Model-based Reinforcement Learning

We model the control task as a finite-horizon Markov decision process (MDP) $\mathcal{M} = (\mathcal{S}, \mathcal{A}, p, r, \gamma, H)$ with state $s \in \mathcal{S}$, action $a \in \mathcal{A}$, transition kernel $p(\cdot \mid s, a)$, reward $r(s, a)$, discount $\gamma \in [0, 1)$, and planning horizon $H$. A policy $\pi(a \mid s)$ seeks to maximize the expected return $\mathbb{E}\left[\sum_{t=0}^{H-1} \gamma^t r(s_t, a_t)\right]$ where $s_{t+1} \sim p(\cdot \mid s_t, a_t)$ and $a_t \sim \pi(\cdot \mid s_t)$.

In model-based reinforcement learning (MBRL), we learn a dynamics model $\tilde{f}$ (and optionally a reward model $\tilde{r}$) from collected data $\mathcal{D}$, and then use $\tilde{f}$ either to (i) train a parameterized policy/value function entirely or partially in imagination ("world-model branch"), or (ii) plan action sequences online at decision time ("pure planning branch") (Ha & Schmidhuber, 2018; Chua et al., 2018; Janner et al., 2019). In this work, we focus on the pure planning branch with receding-horizon model predictive control (MPC) solved by a sampling-based optimizer.

## B.2 MBRL with an MPC policy

Let $\tilde{f}$ denote a learned (possibly stochastic) dynamics model and $\tilde{r}$ a learned or known reward. At each control step $t$, model predictive control (MPC) solves a finite-horizon *open-loop* planning problem over an action sequence $\mathbf{a}_{t:t+H-1} = (a_t, \ldots, a_{t+H-1}) \in \mathcal{A}^H$, executes only the first action, and replans at $t{+}1$. The optimization writes

$$\max_{\mathbf{a}_{t:t+H-1} \in \mathcal{A}^H} \hat{J}_t(\mathbf{a}_{t:t+H-1}) := \mathbb{E}_{\tilde{f}}\left[\sum_{\tau=t}^{t+H-1} \gamma^{\tau-t}\, \tilde{r}(s_\tau, a_\tau)\right] \quad \text{s.t.} \quad s_{\tau+1} \sim \tilde{f}(\cdot \mid s_\tau, a_\tau). \tag{20}$$

Here $\gamma \in [0, 1]$ is a discount factor and the expectation is taken over the model rollout randomness (e.g., process noise and/or sampling across an ensemble). The receding-horizon replanning provides closed-loop feedback and naturally accommodates constraints (e.g., $a_\tau \in \mathcal{A}$) via sampling within bounds, projection, or penalty terms. In modern MBRL, equation 20 is commonly solved by randomized trajectory optimization methods such as random shooting, the cross-entropy method (CEM), and information-theoretic MPPI (Chua et al., 2018; Nagabandi et al., 2018; Williams et al., 2016).

**CEM for trajectory optimization.** CEM maintains a parametric sampling distribution $q_\theta$ over length-$H$ action sequences and iteratively concentrates "action samples" on high-return regions. A standard parameterization in continuous control is a time-indexed diagonal Gaussian,

$$q_\theta(\mathbf{a}_{t:t+H-1}) = \prod_{\ell=0}^{H-1} \mathcal{N}\big(a_{t+\ell}\,;\, \mu_\ell, \mathrm{diag}(\sigma_\ell^2)\big), \qquad \theta = \{\mu_\ell, \sigma_\ell^2\}_{\ell=0}^{H-1}. \tag{21}$$

Starting from $\theta^{(0)}$, each CEM iteration $l = 0, \ldots, L{-}1$ proceeds as follows:

1. **Sampling.** Draw $N$ candidate sequences $\{\mathbf{a}^i\}_{i=1}^N \sim q_{\theta^{(l)}}$, clipping or projecting to $\mathcal{A}$ if needed.

2. **Evaluation.** Estimate model-predicted return $\hat{J}_t(\mathbf{a}^i)$ by rolling out each sequence under $\tilde{f}$ for $H$ steps. With ensembles or particles, average across samples.

3. **Elite selection.** Keep the top-$M$ elites $\mathcal{E}^{(l)} = \arg\mathrm{topM}_i\, \hat{J}_t(\mathbf{a}^i)$ (often $M = \lceil \rho N \rceil$ with elite ratio $\rho \in [0.05, 0.2]$).

4. **Distribution update.** Compute empirical moments from elites and optionally exponentially smooth:

$$\tilde{\mu}_\ell^{(l+1)} = \frac{1}{M} \sum_{\mathbf{a} \in \mathcal{E}^{(l)}} a_{t+\ell}, \qquad\qquad \tilde{\sigma}_\ell^{2\,(l+1)} = \frac{1}{M} \sum_{\mathbf{a} \in \mathcal{E}^{(l)}} \big(a_{t+\ell} - \tilde{\mu}_\ell^{(l+1)}\big)^{\odot 2}, \tag{22}$$

$$\mu_\ell^{(l+1)} = \alpha\, \tilde{\mu}_\ell^{(l+1)} + (1{-}\alpha)\, \mu_\ell^{(l)}, \qquad\qquad \sigma_\ell^{2\,(l+1)} = \beta\, \tilde{\sigma}_\ell^{2\,(l+1)} + (1{-}\beta)\, \sigma_\ell^{2\,(l)}, \tag{23}$$

with smoothing coefficients $\alpha, \beta \in [0, 1]$ and elementwise square, denoted by $\odot 2$. In practice one often clamps variances, $\sigma_\ell^{2\,(l+1)} \leftarrow \mathrm{clip}\big(\sigma_\ell^{2\,(l+1)}, \sigma_{\min}^2, \sigma_{\max}^2\big)$, to prevent premature collapse or excessive exploration.

After $L$ iterations, MPC executes the *first* action of the best sampled sequence and *warm-starts* the next control step by shifting the mean trajectory forward and appending a default tail, e.g.,

$$\mu_\ell^{\mathrm{warm}} \leftarrow \begin{cases} \mu_{\ell+1}, & \ell = 0, \ldots, H-2, \\ a_{\mathrm{tail}}, & \ell = H-1, \end{cases} \qquad \sigma_\ell^{2,\mathrm{warm}} \leftarrow \begin{cases} \sigma_{\ell+1}^2, & \ell = 0, \ldots, H-2, \\ \sigma_0^2, & \ell = H-1, \end{cases}$$

where $a_{\mathrm{tail}}$ (e.g., zero or the last mean action) and $\sigma_0^2$ control exploration at the horizon boundary (see (Chua et al., 2018)). This pure-planning pipeline is simple and effective but can be sensitive to initialization and may lose diversity if $q_\theta$ collapses too quickly.

### B.3 Bregman Centroid–Guided MPC for MBRL

We wrap $\mathcal{BC}$-**EvoCEM** around a standard MPC loop without modifying any worker's internal CEM iterations. At each control step $t$, $K$ workers run CEM *independently* (different random seeds and, optionally, different hyperparameters, as the Decentralized CEM (Zhang et al., 2022)). After they finish, we compute a solution-weighted Bregman centroid across their trajectory distributions and use this centroid to *warm-start* all workers at time $t+1$. This replaces the usual warm-start that shifts the single best sequence (Chua et al., 2018). To avoid consensus collapse, we periodically *respawn* low-relevance workers inside a centroid-centered Bregman trust region.

- **What we $\underline{\text{DO NOT}}$ change.** Within each worker, CEM proceeds exactly as usual (sampling, elite selection, exponential smoothing, variance clamping, etc.). We never enter the worker's inner loop.

- **What we $\underline{\text{DO}}$ change.** Across workers and *between* MPC steps, we (i) aggregate the final worker distributions into a Bregman centroid; (ii) warm-start *all* workers at $t+1$ from the *shifted centroid*; and (iii) optionally respawn a subset of workers inside a centroid-centered Bregman trust region to preserve diversity.

#### B.3.1 Bregman centroid for diagonal Gaussians

Let $\{q_{\theta_k}\}_{k=1}^K$ be the $K$ worker distributions at the end of control step $t$. We form weights $w_k \geq 0$ with $\sum_k w_k = 1$ from the workers' scores (e.g., softmax of best returns). For each time index $\ell$ and each action dimension, the centroid $q_{\boldsymbol{\theta}_c}^\star$ is diagonal Gaussian with parameters

$$\mu_\ell^\star = \sum_{k=1}^K w_k\, \mu_{k,\ell}, \tag{24}$$

$$\sigma_\ell^{\star\,2} = \sum_{k=1}^K w_k\big(\sigma_{k,\ell}^2 + \mu_{k,\ell} \odot \mu_{k,\ell}\big) \;-\; \mu_\ell^\star \odot \mu_\ell^\star, \tag{25}$$

where $\odot$ is the elementwise product. Equivalently, we have the following interpretation

$$\sigma_\ell^{\star\,2} \;=\; \underbrace{\sum_k w_k \sigma_{k,\ell}^2}_{\text{within-worker variance}} \;+\; \underbrace{\sum_k w_k(\mu_{k,\ell} - \mu_\ell^\star) \odot (\mu_{k,\ell} - \mu_\ell^\star)}_{\text{between-worker dispersion}}.$$

The centroid mean in equation 24 is the weighted average of worker means. The centroid variance equation 25 is the sum of (i) the average *within-worker* exploration scale and (ii) the *between-worker* disagreement.

### B.3.2   Centroid-guided warm start across MPC steps

At the end of step $t$, compute $\boldsymbol{\theta}_c$ via equation 24–equation 25. To warm-start *all* workers at $t+1$, shift the centroid forward by one stage (receding horizon) and append a default tail:

$$\mu_\ell^{\text{warm}} \leftarrow \begin{cases} \mu_{\ell+1}^\star, & \ell = 0, \dots, H-2, \\ a_{\text{tail}}, & \ell = H-1, \end{cases} \qquad \sigma_\ell^{\text{warm 2}} \leftarrow \begin{cases} \max(\sigma_{\ell+1}^{\star\,2}, \sigma_{\min}^2), & \ell = 0, \dots, H-2, \\ \sigma_{\text{tail}}^2, & \ell = H-1, \end{cases} \tag{26}$$

where $a_{\text{tail}}$ is a default terminal action (e.g., zeros or the last centroid mean) and $\sigma_{\min}^2$ ensures a variance floor. The same $(\mu_\ell^{\text{warm}}, \sigma_\ell^{\text{warm 2}})$ is given to *every* worker as $\theta_k^{(0)}$ at the next step. Practically, the variance is reset to the initial value to encourage per-worker exploration (Chua et al., 2018). Workers then run their internal CEM loops as usual from this common initialization,

### B.3.3   Weighted aggregation

We compute worker weights $w_k$ from the best sequence return $\hat{J}_k^{\text{best}}$. A robust choice is a tempered softmax:

$$w_k \;=\; \frac{\exp\big((\hat{J}_k^{\text{best}} - \max_j \hat{J}_j^{\text{best}})/\tau\big)}{\sum_{j=1}^K \exp\big((\hat{J}_j^{\text{best}} - \max_i \hat{J}_i^{\text{best}})/\tau\big)}, \qquad \tau > 0, \tag{27}$$

where smaller $\tau$ increases concentration (faster consensus), larger $\tau$ preserves diversity.

### B.3.4   Respawn via trust–region sampling

We respawn low–relevance workers inside the *right-sided* Bregman ball in mean space centered at the warm-shifted centroid $(\mu^{\text{warm}}, \sigma^{\text{warm 2}})$:

$$\widehat{\mathcal{S}}_\Delta := \Big\{ \mu : \; \mathrm{D}_{\Psi^*}\big(\mu \parallel \mu^{\text{warm}}\big) \leq \Delta \Big\}.$$

For the diagonal Gaussian parameterization equation 21 with fixed variances, the mean-space generator is quadratic and

$$\mathrm{D}_{\Psi^*}\big(\mu \parallel \mu^{\text{warm}}\big) = \frac{1}{2} \sum_{\ell=0}^{H-1} \big\|\mu_\ell - \mu_\ell^{\text{warm}}\big\|_{\text{diag}(\sigma_\ell^{\text{warm}\,-2})}^2 \tag{28}$$

$$= \frac{1}{2}\,(\mu - \mu^{\text{warm}})^\top H\,(\mu - \mu^{\text{warm}}), \quad \text{where } H := \text{diag}\big(\sigma^{\text{warm}\,-2}\big), \tag{29}$$

so $\widehat{\mathcal{S}}_\Delta$ is the ellipsoid

$$\widehat{\mathcal{S}}_\Delta = \Big\{ \mu : \; (\mu - \mu^{\text{warm}})^\top H(\mu - \mu^{\text{warm}}) \leq 2\Delta \Big\}.$$

This is exactly the ellipsoidal trust region from Sec. 4, and here the proxy sampler is *exact* for the mean block (quadratic $\Psi$ in $\mu$).

Given the centroid at time $t$ (before shift) with parameters $(\mu^\star, \sigma^{\star\,2})$, compute each worker's score (Sec. B.3.1)

$$\gamma_k \;=\; \mathrm{D}_{\Psi^*}\big(\mu_k \parallel \mu^\star\big) = \frac{1}{2} \sum_{\ell=0}^{H-1} \big\|\mu_{k,\ell} - \mu_\ell^\star\big\|_{\text{diag}(\sigma_\ell^{\star\,-2})}^2.$$

Select a respawn set $\mathcal{R}$ as the top-$r$ fraction by $\gamma_k$.

**Sampling inside the ellipsoid.** Let $D := H \times d_a$ be the total mean dimension. Define the diagonal whitening scales $s := \sigma^{\text{warm}}$ stacked over all times/dimensions and $S := \text{diag}(s)$. Uniform sampling in the ellipsoid is obtained by drawing uniformly in the Euclidean ball and mapping back:

$$y \sim \text{Unif}\big\{y : \|y\|_2 \leq \sqrt{2\Delta}\big\}, \qquad \mu_{\text{new}} = \mu^{\text{warm}} + S\,y,$$

since $(\mu_{\text{new}} - \mu^{\text{warm}})^\top H(\mu_{\text{new}} - \mu^{\text{warm}}) = \|y\|_2^2 \leq 2\Delta$. A convenient implementation is $y = r\,v$ with $v \sim \text{Unif}(\mathbb{S}^{D-1})$ and $r = \sqrt{2\Delta}\,u^{1/D}$, $u \sim \text{Unif}[0,1]$.

---

**Algorithm 5** RESPAWNDIAGONALGAUSSIAN (centroid trust–region sampling)

---

**Require:** Warm centroid $(\mu^{\text{warm}}, \sigma^{\text{warm}\,2})$, radius $\Delta$, respawn set $\mathcal{R}$, action set $\mathcal{A}$ (box)

1: $s \leftarrow \text{vec}(\sigma^{\text{warm}}); \; S \leftarrow \text{diag}(s); \; D \leftarrow H \times d_a$
2: **for** $k \in \mathcal{R}$ **do**
3:      Draw $v \sim \text{Unif}(\mathbb{S}^{D-1})$, $u \sim \text{Unif}[0,1]$, $r \leftarrow \sqrt{2\Delta}\,u^{1/D}$
4:      $\mu_k^{(0)} \leftarrow \mu^{\text{warm}} + S\,(r\,v)$                              $\triangleright$ Uniform in $\{\mu : \mathrm{D}_{\Psi^*}(\mu \| \mu^{\text{warm}}) \leq \Delta\}$
5:      Project each time step to $\mathcal{A}$: $\mu_{k,\ell}^{(0)} \leftarrow \Pi_{\mathcal{A}}\big(\mu_{k,\ell}^{(0)}\big)$
6:      Set variance: $\sigma_k^{2,(0)} \leftarrow \max\big(\sigma^{\text{warm}\,2}, \sigma_{\min}^2\big)$                     $\triangleright$ Clamp elementwise
7: **end for**
8: **return** $\{(\mu_k^{(0)}, \sigma_k^{2,(0)})\}_{k \in \mathcal{R}}$

---

**Practical considerations.** (i) With diagonal, fixed variances, the mean-space generator is quadratic, so Algorithm 5 is *exact*. (ii) A practical calibration for choosing $\Delta$ is to estimate the *information radius*; set and adjust $\Delta$ based on the variance of the relevance scores, which captures the between-worker spread. (iii) *Sampling complexity.* $O(D)$ per respawned worker with one random direction and one diagonal scale.

### B.3.5 Computational overhead and algorithmic advantage

The net extra cost versus vanilla MPC+CEM is *negligible*: computing equation 24–equation 25 is $\mathcal{O}(KHd_a)$ elementwise arithmetic, with no additional model rollouts. The method is a **drop-in wrapper** that leaves inner-loop CEM vectorization intact and is equally compatible with random shooting or MPPI, e.g., simply replace each worker's optimizer, and only the cross-worker centroid/warm-start remains.

# C   (Extended) Technical details

## C.1   Proof of Proposition 1

*Proof.* By linearity of expectation,

$$\sum_i w_i \operatorname{KL}(p_{\theta_i} \| p_\theta) = \sum_i w_i \, \mathbb{E}_{p_{\theta_i}}[\log p_{\theta_i}] - \mathbb{E}_{p_{\mathrm{mix}}}[\log p_\theta] = \operatorname{KL}(p_{\mathrm{mix}} \| p_\theta) + \mathrm{const},$$

where const $= \sum_i w_i \, \mathbb{E}_{p_{\theta_i}}[\log p_{\theta_i}] - \mathbb{E}_{p_{\mathrm{mix}}}[\log p_{\mathrm{mix}}]$ is constant in $\theta$. The second equality follows from $\operatorname{KL}(p_{\mathrm{mix}} \| p_\theta) = -\mathbb{E}_{p_{\mathrm{mix}}}[\log p_\theta] + \mathrm{const}$. $\square$

## C.2   Proof of Corollary 1

*Proof.* Following the Proposition 1, we have

$$\mathbb{E}_{p_{\mathrm{mix}}}[\log p_\theta(X)] = \langle \theta, \, \mu \rangle - \Psi(\theta) + \mathrm{const}, \qquad \text{where } \mu := \mathbb{E}_{p_{\mathrm{mix}}}[T(X)] = \sum_i w_i \eta_i.$$

Maximizing a strictly concave function gives the unique $\boldsymbol{\theta}_c$ with $\nabla\Psi(\boldsymbol{\theta}_c) = \mu$. Using KL-Bregman identity $\operatorname{KL}(p_{\theta_i} \| p_\theta) = \operatorname{D}_\Psi(\theta \| \theta_i)$ (Eq. equation 10), the first-order condition of $\min_\theta \sum_i w_i \operatorname{D}_\Psi(\theta \| \theta_i)$ is the same, so $\boldsymbol{\theta}_c$ is the *left-sided* Bregman centroid in $\theta$. By Legendre duality, $\boldsymbol{\eta}_c$ is the *right-sided* centroid in $\eta$ with generator $\Psi^*$, and $\boldsymbol{\theta}_c = (\nabla\Psi)^{-1}(\boldsymbol{\eta}_c)$ (see Proposition 2 and Remark 1). $\square$

## C.3   Proof of Theorem 1

### C.3.1   Facts

Throughout we work on $\Theta, \mathcal{E} \subset \mathbb{R}^d$ equipped with Lebesgue measure $\lambda^d$. We write $\sigma_{d-1}$ for the surface measure on the unit sphere $\mathbb{S}^{d-1} := \{v \in \mathbb{R}^d \mid \|v\|_2 = 1\}$. The following facts are used (see Villani et al. (2008); Schneider (2013)).

**Fact 1** (Polar coordinates). *Under the polar map $(\rho, v) \mapsto \eta = \eta_c + \rho v$ with $\rho \geq 0$, $v \in \mathbb{S}^{d-1}$, the d-dimensional Lebesgue volume element factorizes as $d\eta = \rho^{d-1} \, d\rho \, d\sigma_{d-1}(v)$.*

**Fact 2** (Uniform distribution). *Let $\rho_{\max} : \mathbb{S}^{d-1} \to (0, \infty)$ be measurable and define*

$$\mathcal{S} := \big\{ \eta_c + \rho v : v \in \mathbb{S}^{d-1}, \ 0 \leq \rho \leq \rho_{\max}(v) \big\}.$$

*Then*

$$\operatorname{Vol}(\mathcal{S}) = \frac{1}{d} \int_{\mathbb{S}^{d-1}} \rho_{\max}(v)^d \, d\sigma_{d-1}(v), \quad \text{and} \quad f_{\mathcal{S}}(\rho \mid v) = \frac{d \, \rho^{d-1}}{\rho_{\max}(v)^d}, \ 0 \leq \rho \leq \rho_{\max}(v).$$

**Fact 3** (Change of variables). *For $\Psi \in C^2(\Theta)$ strictly convex, the gradient map $\nabla\Psi : \Theta \to \mathcal{E}$ is a $C^1$ diffeomorphism with Jacobian $\det \nabla^2\Psi(\theta)$. For any nonnegative $\varphi$,*

$$\int_\Theta \varphi(\theta) \, d\theta = \int_{\mathcal{E}} \varphi\big(\nabla\Psi^{-1}(\eta)\big) \Big| \det \nabla^2\Psi\big(\nabla\Psi^{-1}(\eta)\big) \Big|^{-1} d\eta.$$

*Equivalently, if $f_\eta$ is a density on $\mathcal{E}$ then the pull-back density on $\Theta$ is*

$$f_\theta(\theta) = f_\eta\big(\nabla\Psi(\theta)\big) \Big| \det \nabla^2\Psi(\theta) \Big|.$$

### C.3.2   Auxiliary lemma

We first show the radial Bregman divergence is strictly increasing in the (right-sided) radius.

**Lemma 1** (Monotonicity). *Let $\Psi^*$ be strictly convex and twice differentiable. For fixed $\eta_0$ and $v \in \mathbb{S}^{d-1}$ define*

$$g_v(\rho) := \operatorname{D}_{\Psi^*}\big(\eta_0 + \rho v \, \| \, \eta_0\big), \qquad \rho \geq 0.$$

*Then $g_v$ is strictly increasing on $(0, \infty)$.*

*Proof.* Differentiating yields

$$g_v'(\rho) = \big\langle \nabla\Psi^*(\eta_0 + \rho v) - \nabla\Psi^*(\eta_0),\, v \big\rangle.$$

Strict convexity implies strict monotonicity of $\nabla\Psi^*$. Since $\eta_0 + \rho v \neq \eta_0$ for $\rho > 0$, we have $g_v'(\rho) > 0$. $\qquad\square$

### C.3.3 Main proof

**Theorem 1** (Restatement). *Algorithm 3 produces $\eta_{\mathrm{new}} \sim \mathrm{Unif}\big(\mathcal{S}_\Delta(\eta_c)\big)$ and $\theta_{\mathrm{new}} \in \mathcal{B}_\Delta(\theta_c)$. If $\Psi$ is quadratic, $\theta_{\mathrm{new}}$ is uniformly distributed in $\mathcal{B}_\Delta(\theta_c)$.*

*Proof.* Let $g_v$ be defined as above and set $\rho_{\max}(v)$ by $g_v\big(\rho_{\max}(v)\big) = \Delta$.

**Step 1. Boundary existence & uniqueness.** By Lemma 1, $g_v$ is strictly increasing, so $g_v(\rho) = \Delta$ has a unique root $\rho_{\max}(v) > 0$ for each $v$.

**Step 2. Feasibility.** Algorithm 3 draws $V \sim \mathrm{Unif}(\mathbb{S}^{d-1})$ and $U \sim \mathrm{Unif}[0,1]$, sets $\rho = \rho_{\max}(V)\, U^{1/d}$ and $\eta_{\mathrm{new}} = \eta_c + \rho V$. Because $g_V(\rho) \leq g_V(\rho_{\max}(V)) = \Delta$, $\eta_{\mathrm{new}} \in \mathcal{S}_\Delta(\eta_c)$ and, by duality,

$$\mathrm{D}_\Psi\big(\boldsymbol{\theta}_c \,\|\, \theta_{\mathrm{new}}\big) = \mathrm{D}_{\Psi^*}\big(\eta_{\mathrm{new}} \,\|\, \boldsymbol{\eta}_c\big) \leq \Delta,$$

so $\theta_{\mathrm{new}} \in \mathcal{B}_\Delta(\theta_c)$.

**Step 3. Uniformity in mean space.** Conditioned on $V = v$, $\rho$ has density $d\,\rho^{d-1}/\rho_{\max}(v)^d$ on $[0, \rho_{\max}(v)]$ by construction, which matches Fact 2; integrating over $v$ yields $\eta_{\mathrm{new}} \sim \mathrm{Unif}(\mathcal{S}_\Delta(\eta_c))$.

**Step 4. Pull-back to $\Theta$.** By Fact 3, $f_\theta(\theta) = f_{\mathcal{S}_\Delta}\big(\nabla\Psi(\theta)\big) \big|\det \nabla^2\Psi(\theta)\big|$. For general $\Psi$, $\det \nabla^2\Psi(\theta)$ varies with $\theta$, so $f_\theta$ need not be constant. If $\Psi$ is quadratic, $\nabla^2\Psi$ is constant; hence $f_\theta$ is constant on $\mathcal{B}_\Delta(\theta_c)$, i.e. $\theta_{\mathrm{new}}$ is uniform. $\qquad\square$

# D    Full experimentawl results

## D.1    Benchmark Environment Setup

We follow the evaluation protocol of Zhang et al. (2022) to assess both our method and the baseline algorithms on the suite of robotic benchmarks introduced by Wang et al. (2019); Chua et al. (2018), including classical robotic control problems and high-dimensional locomotion and manipulation problems. Key environment parameters are summarized in Table 5. We refer the interested readers to Zhang et al. (2022) for more details, such as reward function settings, termination conditions, and other implementation specifics. For each case study, all algorithms are trained on three random seeds and evaluated on one unseen seed.

Table 5: Details of Benchmark Environments

| Parameter | Acrobot | CartPole | Hopper | Pendulum | Reacher | Pusher |
|---|---|---|---|---|---|---|
| Train Iterations | 50 | 50 | 50 | 50 | 100 | 100 |
| Task Horizon | 200 | 200 | 1000 | 100 | 50 | 150 |
| Train Seeds | $\{1, 2, 3\}$ | $\{1, 2, 3\}$ | $\{1, 2, 3\}$ | $\{1, 2, 3\}$ | $\{1, 2, 3\}$ | $\{1, 2, 3\}$ |
| Test Seeds | $\{0\}$ | $\{0\}$ | $\{0\}$ | $\{0\}$ | $\{0\}$ | $\{0\}$ |
| Epochs per Test | 3 | 3 | 3 | 3 | 3 | 3 |

## D.2    Algorithms Setup

The key parameters for the proposed $\mathcal{BC}$-**EvoCEM** algorithm and all baseline methods are listed in Table 6. The dynamic model for each benchmark is parameterized as a fully connected neural network: four hidden layers with 200 units each, except for the *Pusher* task, which uses three hidden layers. All algorithms share identical training settings for learning the dynamics model; further details on model learning can be found in Chua et al. (2018) and Zhang et al. (2022).

Table 6: Details of Algorithms (**DE** and **PE**)

| Parameter | PETS-evoCEM | PETS-DecentCEM | PETS-CEM |
|---|---|---|---|
| CEM Type | $\mathcal{BC}$-EvoCEM | DecentCEM | CEM |
| CEM Ensemble Size | 3 | 3 | 1 |
| CEM Population Size | 100 | 100 | 100 |
| CEM Proportion of Elites | 10 % | 10 % | 10 % |
| CEM Initial Variance | 0.1 | 0.1 | 0.1 |
| CEM Internal Iterations | 5 | 5 | 5 |
| Model Learning Rate | 0.001 | 0.001 | 0.001 |
| Warm-up Episodes | 1 | 1 | 1 |
| Planning Horizon | 30 | 30 | 30 |

### D.3 Stochastic models with trajectory sampling

We adopt the probabilistic ensemble with trajectory sampling of Chua et al. (2018), which captures both *epistemic* and *aleatoric* uncertainty. Under this setting, the learning curves of all three optimizers overlap within statistical error (Fig. 8 and Fig. 9). No significant performance gaps emerge. We conjecture that the model's intrinsic stochasticity already supplies enough trajectory dispersion, so additional exploration at the optimizer level offers little incremental benefit. Moreover, the benchmark continuous-control tasks have largely *unimodal* return landscapes, which leaves less room for diversity-seeking mechanisms to improve performance beyond what stochastic trajectory sampling already provides.

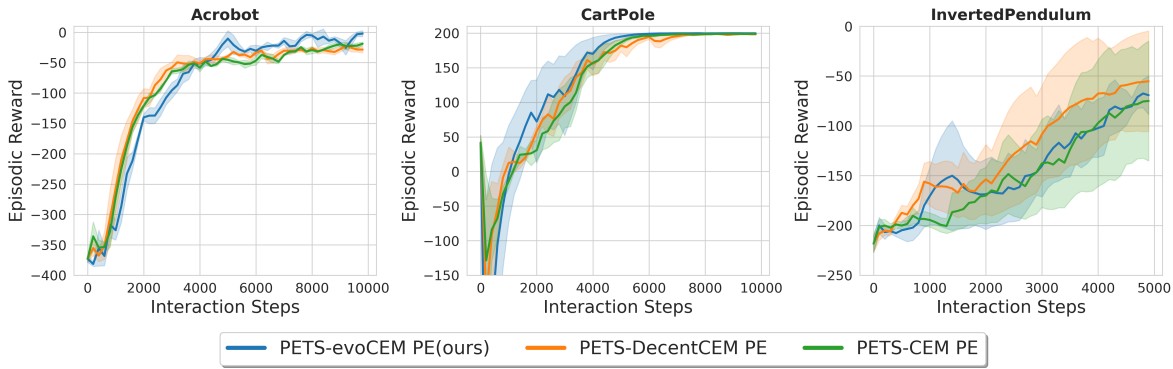

Figure 8: *Training* return curves across 3 control tasks using PETS with different CEM-based optimizers. All methods use the *probabilistic ensemble dynamics model with trajectory sampling* (Chua et al., 2018). Curves show mean performance over 3 random seeds.

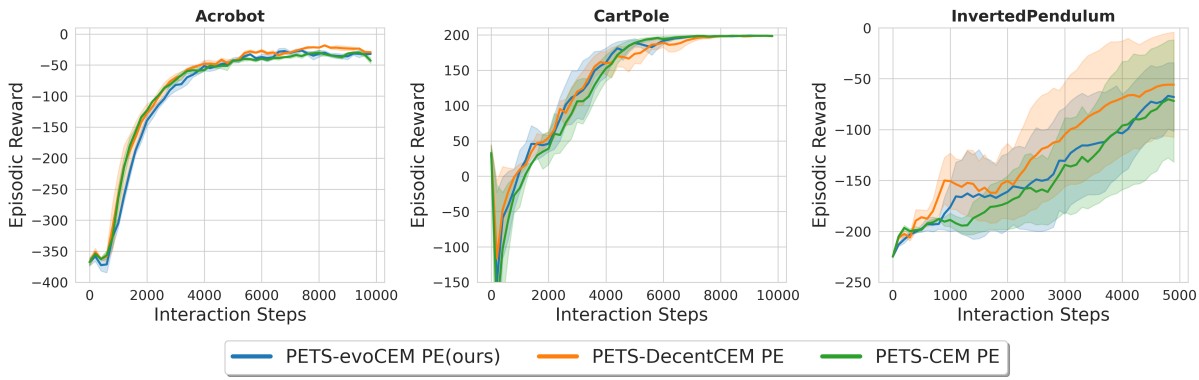

Figure 9: *Testing* return curves across 3 control tasks using PETS with different CEM-based optimizers. All methods use the *probabilistic ensemble dynamics model with trajectory sampling* Chua et al. (2018). Curves show mean performance over 3 random seeds.

### D.4    (NEW) Additional experiments on a public, modular MBRL Library

We perform cross-validation in *MBRL-Lib* (Pineda et al., 2021), a public, modular PyTorch codebase for model-based RL that exposes *interchangeable* components for dynamics, planning, and optimizers. This lets us hold the PETS-style pipeline (Chua et al., 2018) fixed while exchanging only the trajectory optimizer, ensuring an apples-to-apples comparison without confounding changes to data handling, rollout code, or evaluation.

#### D.4.1    Implementations: vanilla CEM, decentralized CEM, and BC-guided CEM

We evaluate three optimizers behind the same MPC/planning interface:

1. **Vanilla CEM** (library default): we use the built-in CEM optimizer as-is.

2. **Decentralized CEM (DecentCEM):** a light wrapper of the built-in CEM that decentralizes the parameter updates.

3. **BC-guided warm-start (ours):** warm-starts **DecentCEM** with Bremgman Centroid of previous solutions and periodically *refresh* the ensemble of CEM workers (see Appdendix B.3 for details).

In all cases, we use the same planer (`TrajectoryOptimizerAgent` in the codebase), reward/termination functions, replay buffer, and evaluation protocol provided by the library. Only the optimizer update rule and initialization are changed.

#### D.4.2    Dynamics model setups and hyperparameters

We compare two standard dynamics configurations in PETS-style pipeline:

**Single Gaussian model (no epistemic).** A single probabilistic MLP with a diagonal-Gaussian head is trained by negative log-likelihood (NLL) on *delta* transitions,

$$\Delta s_t := s_{t+1} - s_t, \qquad p_\theta(\Delta s_t \mid s_t, a_t) = \mathcal{N}\big(\mu_\theta(s_t, a_t), \, \mathrm{diag}\, \sigma_\theta^2(s_t, a_t)\big).$$

The training objective is $\min_\theta \, -\sum_t \log p_\theta(\Delta s_t \mid s_t, a_t)$. At planning time we propagate the predictive mean only,

$$s_{t+1} = s_t + \mu_\theta(s_t, a_t),$$

thereby removing model sampling effects and isolating optimizer behavior.

**Bootstrapped ensemble with TS1 propagation.** An ensemble of $M = 5$ MLPs, each trained on a bootstrap resample with the same Gaussian head and NLL objective. During planning we use PETS-style "TS1" rollouts (Chua et al., 2018): for each candidate trajectory, sample a single ensemble member and roll it out over the horizon, which provides epistemic exploration.

Unless otherwise stated, all models share the same backbone: three hidden layers with 200 units each and SiLU activations. All remaining training and environment hyperparameters follow the MBRL-Lib defaults. See (Pineda et al., 2021) for environment-specific details (e.g., reward overrides, termination conditions, and preprocessing).

#### D.4.3    Environments and return reference ranges

We validate on three environments that have PETS baselines reported in the MBRL-Lib paper, namely, Inverted Pendulum, HalfCheetah, and a continuous version of CartPole. For sanity checks, we compare our absolute returns to the PETS ranges reported for these tasks in the MBRL-Lib paper (Pineda et al., 2021). When using the ensemble+TS1 regime, our absolute returns fall within the reported bands (within seed variance), while the deterministic Single-Gaussian regime yields lower absolute returns as expected.

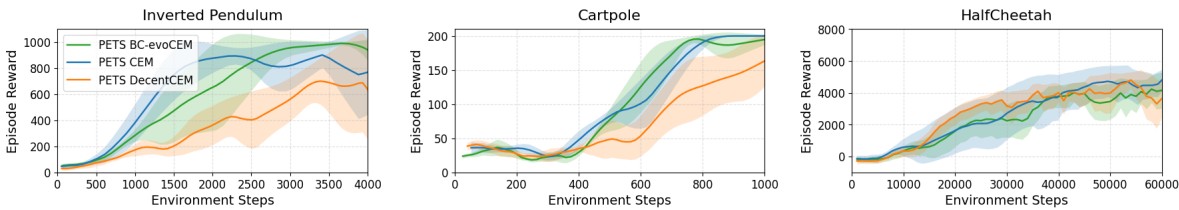

Figure 10: Training return curves across 3 control tasks using PETS with different CEM-based optimizers. All methods use a single Gaussian MLP dynamics model. Curves represent the mean performance across three random seeds.

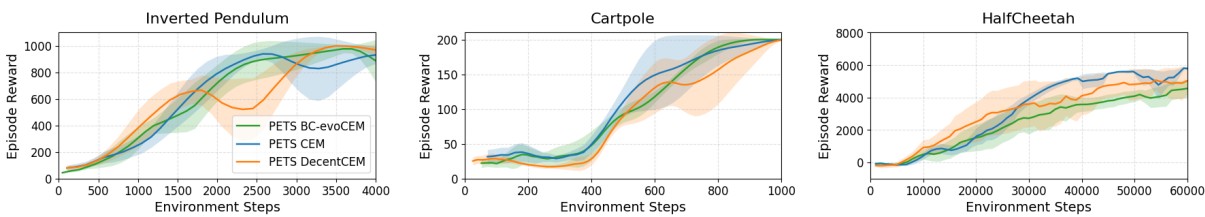

Figure 11: Training return curves across 3 control tasks using PETS with different CEM-based optimizers. All methods use a probabilistic ensemble model with Gaussian heads and the TS1 sampling strategy. Curves represent the mean performance across three random seeds.

### D.4.4   Results

Figures 10 and 11 report learning curves for three PETS planners under two dynamics regimes: a single Gaussian MLP model and an ensemble with TS1 uncertainty propagations. Across tasks, the qualitative trends are consistent with prior observations: *optimizer-level exploration* and *explicit uncertainty-handling* are complementary when model capacity is limited, whereas stronger epistemic modeling reduces optimizer-level differences.

**Single-model dynamics (Gaussian MLP).**   When planning with a single MLP dynamics model (Fig. 10, our BC variant attains faster learning and higher or comparable asymptotic returns on the low-dimensional tasks (Inverted Pendulum, Cartpole), with noticeably tighter confidence bands than the baselines. In this regime, optimizer-induced diversity acts as a surrogate for missing epistemic coverage when epistemic uncertainty is not represented by the model itself. In contrast, DecentCEM tends to underperform given a fixed population budget: partitioning the population across workers reduces per-worker effective coverage of the high-variance trajectory objective, and the "best-worker" selection rule amplifies the model bias, leading to premature convergence or high variance.

**Ensemble dynamics with TS1 sampling.**   With an uncertainty-aware dynamics ensemble and TS1 rollouts (Fig. 11), the performance gap among planners narrows on Pendulum and Cartpole. Because each rollout is conditioned on a different model draw, the planning process already injects structured exploration over epistemic hypotheses. Consequently, optimizer design plays a less dominant role. Nonetheless, our BC-variant remains competitive while exhibiting lower inter-seed variance relative to DecentCEM, indicating that the trust region suppresses the large "modeling–stochasticity" interactions that arise when different ensemble members extrapolate optimistically.

**High-dimensional task (HalfCheetah).**   The relative advantage of our method is reduced on the HalfCheetah task. We hypothesize that this is largely a *trust-region scaling* issue: our experiments used

a single, fixed trust-region budget across tasks. For high-dimensional action sequences and long horizons, a constant trajectory-level constraint can be overly conservative, effectively shrinking the per-dimension step size. A principled remedy is to scale the trust-region radius with problem/task size (e.g., proportional to $\sqrt{H, d_a}$, where $H$ is the planning horizon and $d_a$ the action dimension), or to impose per-time-step constraints.

**Observations on pure decentralization.** Pure decentralization (DecentCEM) under a fixed global sampling budget generally underperforms in both dynamics regimes. Without coupling mechanisms such as warm starts from a learned policy prior (see the policy initialization strategy in (Zhang et al., 2022)), independent workers explore largely disjoint regions with limited elite exchange. This reduces the statistical reliability of elite estimates within each subpopulation and, combined with best-worker selection, amplifies the model bias. The effect is most pronounced with a single-model dynamics, where optimizer noise and model misspecification interact adversely.

In summary, these findings are consistent with prior experimental results and provide additional insights

- Under model capacity limitations, optimizer-level uncertainty handling (BC and its induced trust-region) and exploration mechanisms provide gains in sample efficiency and stability. When epistemic uncertainty is represented explicitly by an ensemble with TS1 sampling, the planners' performance is marginal (still, trust-region methods reduce return variance).

- For high-dimensional tasks, trust-region budgets must scale with horizon and dimensionality. Otherwise, conservative updates can mask the benefits of uncertainty-aware planning.

- Decentralization should be paired with lightweight coupling or warming strategies. Otherwise, splitting a fixed sampling budget across workers is statistically inefficient and prone to myopic selection.

These findings explain the observed patterns across Figures 10–11 and point to concrete design choices such as budget scaling, mild coupling between workers, and model–optimizer co-design.

