# OpenReview forum: "Bregman Centroid Guided Cross-Entropy Method"
_TMLR — Rejected by TMLR_

### Review · Reviewer_MoNT · 2025-08-26

**Summary Of Contributions:**

This paper proposes an extension of the Cross-Entropy Method (CEM) by introducing a new sampling strategy based on Bregman divergence. The approach involves selecting a centroid by minimizing the Bregman divergence across samples, then scoring samples according to their divergence from this centroid. Low-scoring samples are resampled within a trust region. The method is evaluated on standard benchmarks as well as one real-world example.

**Audience:**

Yes

**Audience Explanation:**

The topic is interesting and relevant for the TMLR audience, particularly those working on optimization and reinforcement learning.

**Claims And Evidence:**

No

**Claims Explanation:**

The optimization scheme relies heavily on heuristic choices. It is not clear why Bregman divergence is an appropriate metric in this context, nor why the proposed resampling procedure centered on the divergence-based centroid offers a theoretical advantage over standard CEM or its variants. Moreover, the experimental results are not aligned with concurrent literature, raising concerns about the validity of the claims. I elaborate on these issues in my requested changes below.

**Requested Changes:**

Criteria for Acceptance:
1) The experimental results are not convincing when compared to concurrent literature. For example:
Your reported results on the inverted pendulum environment (reward ≈ –50) are significantly worse than those in “A Simple Decentralized Cross-Entropy Method” (DecentCEM), which achieves reward ≈ 0.
The results table is incomplete, as several environments commonly used in the literature are missing.
A clearer and more comprehensive comparison with concurrent baselines is essential for acceptance.


2) Please provide a theoretical justification for why the proposed algorithm—particularly the use of Bregman divergence and the centroid-based sampling scheme—should be expected to outperform or improve upon standard (including decentralized) CEM.
If a theoretical explanation is not feasible, I strongly encourage a broader ablation study comparing your approach to alternative divergence measures. For instance:

- Centroid-like samples could also be constructed via kernel herding (Maximum Mean Discrepancy) or by optimizing the Kernel Stein Discrepancy.

- Trust-region samplers can also be heuristically constructed using methods such as covariance adaptation (e.g., CMA-ES), KL-divergence–based trust regions (as in TRPO), or entropy-based resampling.

Minor Comments:

Figure 1 lacks clarity. In particular, the comparison between subfigures (a) and (b) is not well explained. Please expand the caption or text to clarify what the reader is meant to observe.

---

> ### Author Response · Authors · 2025-10-11
> **Provide theoretical justification and more experimental results**
>
> We thank the reviewer for the thoughtful feedback. Below we address the concerns point‑by‑point.
>
> ---
>
> ### On the comments of ''heuristic choice''
>
> The choices are **NOT heuristics**, but are consequences of exponential family(EF)-Bregman geometry.
>
> A fundamental result of EF-Bregman is that every regular EF member with a cumulant $\Psi$ corresponds to an *unique* Bregman geometry via the conjugate $\Psi^*$(or $\Psi$ by duality); we now cite this result and state its consequences *explicitly* in Preliminaries.
>
> Sec.6 now shows that the proposed centroid/divergence are not *ad‑hoc*: given that CEM can seen as a KL-projection onto the parametric family, the best single EF representative of the workers’ mixture is the Bregman centroid (see Prop. 1 and Cor. 1). Thus the centroid arises as the KL projection, which yields moment matching in closed form. The introduced *Relevance Score* is the likelihood difference at the centroid, and the Bregman ball *Trust-Region* can seen as the likelihood level sets. These results make the scoring and sampling rules consequences, not heuristics (see Appendix A.1–A.3).
>
> ---
>
> ### Requested change
>
> 1. > Results not aligned with concurrent literature.
>
>     **Clarification on the perceived mismatch** PETS has two decoupled modules:
>     - (i) a dynamics model for rollouts and
>     - (ii) a trajectory optimizer for MPC.
>
>     In our main text we **intentionally isolate the optimizer** by holding the PETS pipeline fixed and using deterministic dynamics models. This removes model‑induced stochasticity so any performance differences arise from the optimizer alone. By contrast, note that the concurrent literature, DecentCEM, reports results with *(i)probabilistic ensemble models* and *(ii)ensemble of parameterized policy together with automatic differentiation through the dynamics model to find warm-start for each CEM worker.* Under those settings, **absolute returns** are higher because the model and policy network provide stronger uncertainty handling. This is a modeling and differentiation through the model effect, NOT evidence that the optimizer alone is superior.
>
>     We state this design choice explicitly in Sec. 7.3 and emphasize that our claims are within‑pipeline, **one‑to‑one** comparisons of optimizers. When we switch to the probabilistic ensemble regime used by concurrent work, absolute returns increase and optimizer gaps narrow, exactly as our Appendix D.3 shows.
>
>
> 2. > Need more experiments.
>
>     To address the concern regarding comparability with concurrent literature and missing envs, we re‑implemented and cross‑validated three optimizers (CEM, DecentCEM, and our BC‑guided CEM) within a public, third‑party modular codebase[1]. MBRL‑Lib exposes modules for dynamics, planning, and optimizers, making it suited for *controlled, like‑for‑like comparisons of optimizers inside the same pipeline.* We only wrap MBRL‑Lib’s existing CEM into a decentralized and our BC-guided update scheme. No other code paths are changed. For each optimizer, we ran experiments on two model regimes (single Gaussian and probabilistic ensemble with uncertainty propagation) on **enviroments where MBRL‑Lib’s paper reported absolute returns ranges.**  See App. D4 for additional results and discussions.
>
>       [1] Pineda, Luis, et al. "Mbrl-lib: A modular library for model-based reinforcement learning." arXiv preprint arXiv:2104.10159 (2021).
>
> 2. >Improvement over vanilla/decentralized CEM
>
>     Targeted use of sampling-budget. Consider a fixed rollout budget B per iteration. Vanilla CEM spends all B samples on a single, unimodal distribution; decentralized CEM splits B across K independent workers with no communication; our method also splits B across K workers but adds a light, information‑geometric coupling after each inner CEM step. With the same total rollout budget, vanilla CEM concentrates too early, and decentralized CEM wastes samples on redundant or poor workers. BC‑EvoCEM turns those wasted samples into targeted exploration by (i) geometric-consistent aggregation, (ii) likelihood‑based relevance, and (iii) trust‑region respawn. The empirical evidence of this budget efficiency is explicitly shown in Fig.4 and furthur demonstrated in rest of experimental results, including the real-world deployment.
>
>
> 4. > Theoretical justification & other alternatives
>
>     Theoretical justification is now included.
>
>     App. A.1 now provides a detailed technical development of BC-evoCEM by showing the close connections between CEM, EF, and Bregman geometry.
>
>     Sec. 6 formalizes the use of Bregman divergence. We also added a structured discussion explaining why other alternatives (e.g., barycenters) lack one or more of these properties in this setting.
>
> 5. > Figure 1 caption
>
>     Revised for clarity.

---

### Review · Reviewer_TuVU · 2025-09-01

**Summary Of Contributions:**

Authors propose a new trajectory "optimizer" for model-based reinforcement learning. Authors suggest by incorporating a scoring step and a sampling step that incorporate Bregman divergence as a norm, it can continue benefits of both ensemble method and decentralized method.

**Audience:**

Yes

**Audience Explanation:**

N/A

**Claims And Evidence:**

No

**Claims Explanation:**

To be totally honest, I feel the narrative of the submitted manuscript is largely unclear.

- Regarding the problem authors try to solve, I as a reader would like to understand the _exact_ formulation of the problem authors try to solve. Please, state the problem very clearly.

- When it comes to reinforcement learning problems, I want the context to be very clearly explained (in main text or in appendix). What are the states, actions, and policies exactly? Explain them.

- In algorithm 1, what does `Distributed CEM` do exactly?

- Why is $\nabla \Psi$ a bijection? Please cite the entire result that justifies this claim.

- In algorithm 4, what is $\tilde{f}$?

- Where is the trust-region sampling method used? If authors are talking about policy gradient method, explain the setup clearly.

- If talking about exponential family, then explain how is it related to any concrete problem this manuscript covers.


In my opinion, this manuscript is not much readable in its current status.

**Requested Changes:**

In my opinion, the entire manuscript should be fundamentally overhauled. Specifically, the problem setup should be made crystal clear, and the results should be clearly stated: what is newly proposed, how each part works together.

---

> ### Author Response · Authors · 2025-10-11
> **Major revision for clarity**
>
> We appreciate the reviewer's emphasis on clarity. We **substantially** revised the paper to state the problem precisely, define the RL/MPC setup, and spell out each algorithmic step.
>
> ---
> ### Requested change
>
> 1. >Problem statement
>
>     We now state the optimization problem explicitly and early. The newly added Section 3 formalizes the black‑box minimization objective and reviews how CEM solves this problem via elite selection and maximum‑likelihood updates (Eq. (5)). This places our contribution explicitly: an ensemble upgrade to CEM.
>
> 2. >MBRL setting
>
>     We added a self‑contained RL background. Appendix B.1-B.2 now defines the MDP, reviews MBRL, and describes the MPC we use in MBRL: at each decision step, we plan an open‑loop action sequence (solved by CEM), execute only the first action, observe the next state, and replan (Eq. (18)). We also give the precise trajectory‑distribution parameterization used by CEM optimizer in continuous control and show their update rules (Eq. (20-21)).
>
>     **IMPORTANT CLARIFICATION:** We are NOT doing any policy‑gradients. The “policy” is the standard MPC policy by planning with CEM, i.e., stochastic trajectory optimization inside a *learned* transition model $\tilde f$.
>
>
> 3. >On the `DistributedCEM`
>
>     Alg.1 uses `DistributedCEM` as a short name for running n independent, standard CEM workers in parallel, each performing the usual CEM inner loop. We added additional comments in Alg.1 for clarity.
>
> 4. >Why $\nabla \Psi$ a bijection?
>
>     In regular, minimal exponential families (EFs) the cumulant $\Psi$ is strictly convex on an open natural‑parameter set $\Theta$. Consequently, the gradient map $\eta=\nabla \Psi(\theta)$ and $\theta = \nabla \Psi^*(\eta)$ is a Legendre bijection between the natural parameter $\theta$ and the mean parameter $\eta$. We cite standard sources and restate this in Sec.2 (Preliminaries) and Appendix A.1, including the EF–Bregman duality used throughout.
>
> 5. >In Alg.4, what is $\tilde f$?
>
>     $\tilde f:= \tilde f(s_t, a_t)$ is the learned (dynamics) transition model in MBRL. We state this explicitly in Sec.5 and A.1 now.
>
> 6. >Where is the trust‑region sampling method used? Is this about policy gradient?"
>
>     Trust‑region sampling is part of BC‑EvoCEM’s ensemble evolution, NOT policy gradient. In Alg. 1, Step 7 we respawn the lowest‑relevance worker by sampling new worker parameters inside a Bregman trust region centered at the ensemble centroid. In the MPC variant (Algorithm 2), we apply the same mechanism keep CEM workers diverse between control steps. Full details (including samplers and the diagonal‑Gaussian specialization used in RL (Alg. 5)) now appear in Sec. 4 and Appendix A.3–A.4 / B.3.4.
>
> 7. >If you invoke exponential families, explain how they relate to a concrete problem here.
>
>     The EF usage is both concrete and explicit: the sampling distributions inside CEM are chosen from regular exponential families, and in our RL experiments they are time‑indexed diagonal Gaussians, an essential EF member. This lets us compute, in closed form:
>     - the Bregman centroid (averaging sufficient statistics),
>     - the relevance scores (as Bregman divergences/log‑likelihood), and
>     - an efficient trust‑region sampler (ellipsoids in mean space).
>
>     Section 4 provides the EF machinery used by BC‑EvoCEM, and Section 4.1 now gives a fully worked, step-by-step Gaussian example. In the MPC/RL setting, Appendix B.3.1–B.3.4 specializes all formulas to the actual diagonal‑Gaussian trajectory distributions we use, including a closed‑form centroid across workers (Eq. (22)–(23)) and the ellipsoidal trust region for respawn (Eq. (26)–(27), Alg. 5).
>
> 8. >What exactly is newly proposed and how do the parts work together?
>
>     We summarize the contribution early and explicitly at the end Sec.1
>     - *Newly proposed:* BC‑EvoCEM, a lightweight, ensemble upgrade to CEM that has principled information aggregation via Bregman centroid and diversity-driven exploration with minimal computation overhead.
>
>     **How parts fit?** Alg. 1 gives the per‑iteration loop: Independent CEM updates → centroid + scoring → Trust‑region respawn (pp. 5–6). Alg. 2 shows the drop‑in MPC wrapper for MBRL: the centroid warm‑starts all workers at the next step, and respawn preserves diversity (pp. 6–7). The EF-Bregman grounding in Sec.6 (Prop. 1, Cor. 1) explains why the centroid is the right aggregate.
>
> ---
> ### Comments on readability
> We substantially revised and reorganized the paper, added problem statement/the RL-MPC appendix, worked examples with a closed‑form table (Tab.1), and explicit algorithms(Algs. 1–5). We also added “Why Bregman?” (Sec.6) to motivate the choice. We hope this addresses your concerns and makes the contribution and presentation clear.

---

### Review · Reviewer_EHD2 · 2025-10-02

**Summary Of Contributions:**

This work seeks to improve the Cross-Entropy Method (CEM) in model-based reinforcement learning by developing strategies to help "elite" samples explore more diverse low-cost regions of the model space (rather than becoming concentrated on a single area as in conventional CEM). The main approach is to use Bregman divergence to define a computationally inexpensive method of ranking the score of samples in terms of their cost and diversity. This score is then used to replace low score samples with new samples selected within a trust region defined by local Bregman geometry which should be reasonably low cost and diverse compared to the existing samples. Due to the properties of exponential family potentials, the score can be calculated using quantities already available in CEM. The dual space geometry of the Bregman ball with a quadratic potential allows efficient sampling within the trust region by simple draws from uniform distributions. The proposed method can be integrated with CEM with little additional cost. Experiments show that the proposed method generally convergence faster than decentralized CEM.

Strengths:
* The paper addresses a relevant and important problem in MBRL. The proposed method is scalable and integrates well with existing approaches.
* The paper has a unified and theoretically sound approach based on Bregman geometry. This perspective both provides a way to rank the usefulness of CEM samples in terms of diversity and cost, along with a way to generate new samples to replace low utility ones.
* The experiments consistently show increased convergence speed compared to decentralized CEM.

Weaknesses
* The proposed method remains limited in its abilities to explore highly multimodal geometries because the Bregman trust region remains a convex and Gaussian-like region. While the method seems useful for spanning a larger convex region, it appears to still relies on decentralized CEM copies to enhance multimodal exploration.

**Audience:**

Yes

**Audience Explanation:**

CEM is a popular method for MBRL and computationally efficient improvements such as the one in this work have the potential to be widely adopted.

**Broader Impact Concerns:**

Broader impacts were not addressed, but this does not impact my assessment of the paper.

**Claims And Evidence:**

Yes

**Claims Explanation:**

The paper is mathematically sound. The claims related to Bregman divergence, including the calculation of the Bregman centroid using sample statistics and the method for generating samples uniformly within the Bregman trust region, are mathematically correct. A variety of MBRL benchmarks are evaluated to show that the proposed method outperforms relevant competitors, in particular decentralized CEM. The claims about computational efficiency appear correct.

**Requested Changes:**

It would be helpful to state what the potential $\Psi$ and sufficient statistic $T$ are explicitly in each of the experimental cases.

---

> ### Author Response · Authors · 2025-10-11
> **Clarify scope and provide EF-Bregman details**
>
> Thank you for the thoughtful and positive evaluation, and for recognizing the scalability, theoretical grounding, and empirical gains of our approach.
>
> ---
>
> ### On the limitation
> We agree that a single Bregman trust region is convex. Our method is designed to improve **local** exploration efficiency and is complementary to decentralization/restarts.
>
> - We now make this scope more explicit in the Limitation section
>
> - Add a brief discussion of a practical *multi-center* variant (cluster elites and apply trust-region sampling per cluster), which acts like a mixture of Bregman balls. A full study is left for future work.
>
> ---
>
> ### Requested change
>  > Make potentials & sufficient statistics explicit
>
> We now added fully worked, step-by-step Gaussian example in Sec 4.1, with all relevant quantities (including potentials, cumulant, mean/natural parameters of EF, etc) summarized in Tab. 1. We also provided a similar walkthrough of how the proposed method integrated into the MPC/MBRL pipeline in Appendix B, where the time-indexed diagonal Gaussian is used as the CEM parameterization.
>
>
> ---
>
> ### Summary
> We appreciate the clear, constructive feedback. We will make the relevant EF-Bregman quantities explicitly in concrete examples, clarify the intended scope and limitations, and refine the presentation accordingly.

---

### Author Response · Authors · 2025-10-24
**Inquiry regarding further reviewer feedback**

Dear reviewers,

We would like to kindly ask whether any further comments or requested changes are expected on our submission. We have addressed available reviews point-by-point and are happy to provide additional explanations or results.

We appreciate your time and effort, and wish to ensure we do not miss any opportunity to clarify our work before the decision phase.

---

### Decision · Action_Editor_RSht · 2025-11-20

**Recommendation:** Reject

**Audience:**

Yes

**Audience Explanation:**

.

**Claims And Evidence:**

No

**Claims Explanation:**

Here are my takeaways of the reviewers' reviews:
- Reviewer EHD2 recommends acceptance, emphasizing the elegance and potential utility of the approach.
- Reviewer MoNT is leaning reject, citing unconvincing results and limited practical gain.
- Reviewer TuVU recommends rejection, pointing to weak empirical evidence and insufficient methodological transparency.

Despite the commendation disparity, I believe they agree overall on the strengths and the weaknesses of the submission. Now that it comes to make a decision, I am particularly concerned about the methodological and experimental rigor and clarity, I therefore concur with the recommendation to reject.

While the paper will not be accepted in its current form, I encourage the authors to continue developing this line of research. Specifically:
- Strengthen empirical evidence: Include repeated trials, variance reporting, and comparisons on standardized public MBRL benchmarks (e.g., dm-control, Meta-World).
- Clarify replicability: Provide full algorithmic hyperparameters and open-source code to facilitate independent verification.
- Refine positioning: Explicitly articulate how BC-EvoCEM differs from and improves upon other ensemble-based or diversity-preserving optimizers.

With more comprehensive experiments and clearer justification of its empirical value, this work could eventually make a strong contribution to the optimization and MBRL communities.

Overall, the technical clarity and reproducibility of this work should be fundamentally overhauled for improved readability.

Addendum:
Please find below the extended concerns as expressed in the final recommendations of the reviewers:
- The main algorithmic novel claim is to replace of first order optimisation method to refresh parameter estimate with a second-order trust-region based optimisation method. I agree with another reviewer that the numerical evidence does not support the usefulness of this optimisation approach, as authors did not report if the presented results are the round that favours authors' argument or are of multiple independent trial averages.
- Implementation details are incomplete: for instance, what are actual $\Psi$ in algorithm 3 implemented in each experiment? In the MBRL setting, how is the environment dynamic function $\tilde{f}$ trained?
- If there is an "aggregation step" to process quantities in the ensemble, what is the $F$ used in each experiment setting? In general, the role of cross-entropy method in MBRL is not well explained. Which part of MBRL does cross-entropy method help?
- The results are not convincing, especially when checking App D. The method introduces a lot of complexity, while not providing any measurable benefit on a public, modular MBRL Library.

**Resubmission Of Major Revision:**

The authors may consider submitting a major revision at a later time.